

# Enhanced approach to match damage-equivalent loads in rotor blade fatigue testing

David Melcher[1], Sergei Semenov[2], Peter Berring[2], Kim Branner[2], and Enno Petersen[1]

[1]Department of Rotor Blades, Fraunhofer IWES, Fraunhofer Institute for Wind Energy Systems, Am Seedeich 45, 27572 Bremerhaven, Germany
[2]Technical University of Denmark, Department of Wind and Energy Systems, Risø Campus, 4000 Roskilde, Denmark

**Correspondence:** David Melcher (david.melcher@iwes.fraunhofer.de)

**Abstract.**

In the design process of current wind turbine blades a critical step is the certification testing to confirm design assumptions and requirements. To demonstrate reliability in fatigue testing the blade shall be loaded in all areas of interest to the load levels, which at the end of such test campaign adequately represent the blade lifetime. These loads are typically derived from

aero-elastic load calculations with a combination of different design load cases in the form of accumulated bending moment distributions. The current practice includes two fatigue test sequences, which are aligned with first flapwise and lead-lag modes respectively with the aim to reach defined target bending moment distributions. These two test sequences combined may not cover all areas of interests and some areas could be tested insufficiently. Also in some areas the conventional target bending moment formulation does not represent fatigue damage of the material correctly, as it is not derived from a stresses

or strain based damage calculation and does not allow for mean load correction. The aim of this work is to demonstrate these shortcomings on a particular test case and propose an enhanced method to derive representative target loads, which cover all areas of interest and are strain proportional, allowing for correct material damage accumulation and mean load correction. It is shown for the test case that compared to the conventional methods the enhanced target loads require 16% higher test loads at certain positions along the blade within the four main load directions and even more for load directions in between.

## 1 Introduction

The design and certification processes of wind turbine rotor blades are essential for ensuring their operational reliability and performance over a lifespan typically ranging from 20 to 30 years. A critical and time consuming component of the certification is fatigue testing of first manufactured instances of a new blade type, which is aimed at validating design assumptions and ensuring that blades can endure the fatigue loads encountered throughout their operational life. As current blades are designed

closer to the limits of the materials and thus have less reserves to resist overloading than older generations, representative fatigue testing gains more importance. These tests subject the blades to cyclic loading conditions derived from a collection of design load cases, primarily based on bending moment distributions, which are combined to represent the blade lifetime.

Nowadays fatigue test campaigns are mostly executed according to the current IEC 61400-23:2014 and DNV ST-0376:2024 standards and typically consist of two consecutive test sequences in flapwise and lead-lag direction. Each fatigue test involves



mounting the blade root to a test block and exciting the blade in resonance at or near its corresponding natural frequency for a defined number of cycles. Test loads are introduced along the blade which need to match or exceed the required target loads. To adjust the load distribution along the blade, additional masses are attached to the blade by using load frames.

The target loads are derived from transient aero-elastic load simulations considering different operational conditions and Design Load Cases (DLCs) of the wind turbine (IEC 61400-1:2019; DNV-ST-0437:2024). IEC 61400-5:2020 (section 6.6.2.2)

specifies to generally use strain proportional loads, but allows to use bending moments as well. Therefore, typically the simulated timeseries are evaluated and accumulated, resulting in target bending moment distributions.

This approach with test sequences in separated loading directions, while established, may not adequately cover all critical areas of the blade. As only the main flapwise and lead-lag directions are loaded and compared to the target loads, the regions in between are not examined and are at risk of under-testing.

Furthermore, conventional target bending moment formulations may not accurately represent material fatigue damage, as they do not consider stress or strain distributions and neglect the influence of mean loads on fatigue behavior. This leads to fatigue testing procedures misrepresenting the fatigue damage, even in the four main directions of the blade.

Both DNV and IEC are continuously working on their standards improvement. DNV published a new version of DNV ST-0376:2024 in April 2024 and IEC committee TC 88/MT 23 is currently working on second revision of the IEC 61400-23:2014

with a forecast for release date in June 2026.

DNV ST-0376:2024 requires to include the calculation method for the theoretical fatigue damage evaluation in the blade test specification and to use an equivalent load amplitude whose associated fatigue damage is equal to the fatigue damage calculated from the design load spectrum to obtain test loads. The draft of the IEC 61400-23:2026 (CD) calls for the tests to be designed for fatigue damage in contrast to the current standard IEC 61400-23:2014 which uses "fatigue damage equivalent loads" as

a test design criterion. These developments show the importance of advancing fatigue testing to achieve more representative loading.

One of the first attempts to load a wind turbine blade more realistically was made by van Delft et al. (1988). They used two slanted hydraulic actuators to apply biaxial bending moments simultaneously, which were derived from real wind speed time series. The next known biaxial test campaign was performed by Hughes et al. (1999) with forced excitation via a bell-crank

mechanism. Such forced excitation approaches are widely used in the aerospace and car industry. Although they can produce the most realistic loading as well as damage initiation and development, they quickly became unfeasible for wind turbine blades due to the size of equipment and energy required for excitation. Therefore further development of test methods was focused on partially or fully utilizing resonance of the system for both uniaxial and biaxial excitation at controlled and phase-locked frequency ratios (e.g. 1:1, 1:2) or at arbitrary frequency ratios resulting from the systems natural frequencies (see White, 2004;

White et al., 2005, 2011; Bürkner and van Wingerde, 2011; Greaves et al., 2012; Greaves, 2013; Snowberg et al., 2014; Post and Bürkner, 2016; Melcher et al., 2020a, b, c; Bürkner, 2020; Castro et al., 2021a, 2022, 2024). Most of these works focused on the testing method and its practical application, while still using conventional bending moment based approaches to derive target loads. Melcher et al. (2020b) used sectorial bending moment based target loads in 30° steps for designing biaxial fatigue tests but still allowed under-loading for the sectors between the main directions. Sectorial equivalent fatigue loads based on transfer





functions were used by Previtali and Eyb (2021) as well. Greaves et al. (2012); Greaves (2013) used strain-based methods and included mean load correction for multiple points along the circumference for the test evaluation, but only considered loads in the main directions as target loads. Freebury and Musial (2000) and Ma et al. (2018) proposed a way to incorporate mean load corrections into target bending moment derivation. Castro et al. (2021b, 2024) proposed a bending-moment-based but strain-proportional method of deriving target loads for biaxial testing including any desired load direction. Though, they did 65 not include strain-proportional mean load consideration. Summing up, some proposed test methods became unfeasible, and the publications on equivalent target strains were each missing certain important aspects.

Therefore, the current work proposes an approach for deriving target loads which are covering all loading directions. The derived target loads are proportional to strains and have a possibility to perform mean load correction by combining corresponding methods. This study aims to demonstrate the effect of considering the strain based fatigue behaviour and taking the 70 mean load influence into account. This is done on a specific test case to show potential improvements of the conventional target loads to better represent the material fatigue behaviour. An enhanced method based on the work of Castro et al. (2021b, 2024) is proposed for deriving these representative target loads. The proposed approach emphasizes strain proportionality, facilitating accurate material damage accumulation and enabling mean load corrections. As these loads can be derived for any direction, also target loads for biaxial testing can be derived. The derived target loads provide the option to be converted into strain- 75 s/stresses directly or after using Rainflow counting and/or damage accumulation, allowing for correct utilization of the used methods. As the proposed approach can be used for any load direction, it enables target loads for any fatigue test method including biaxial testing. In light of these considerations, this work seeks to refine the understanding of wind turbine blade fatigue testing methodologies and aims to enhance the safety and reliability of blades through improved post-processing of aero-elastic simulations and testing practices. The proposed enhancements of strain proportionality and mean load correction 80 are expected to set a new standard for future certification processes.

## 2   Data processing methods to derive target loads

As every Original Equipment Manufacturer (OEM) has different procedures to derive their target loads for rotor blade fatigue tests and there is no exact procedure in the standards described, here a conventional procedure is assumed. The different processing procedures which are described in this work are visualized as flow diagram in Fig. 1. Processing path 0 is the 85 minimum procedure necessary to derive Damage Equivalent Loads (DEL) in the main directions of the blade. However, it is not recommended as is does not take into account any stiffness properties of the blade sections. Processing path 1 resulting in $M_{\beta,\mathrm{DEL}}$ represents the assumed conventional procedure. The results from processing paths 3.1 and 3.2 are used here as a reference case because they best represent the actual material fatigue behavior. Processing path 2.1 describes the procedure proposed by Castro et al. (2021b), and processing path 2.2 describes the enhanced approach proposed in this work.

To evaluate these target load distribution for a rotor blade fatigue test the procedure described in the following sections is followed.



**Figure 1.** Flow diagram of procedures for processing of load time series resulting in alternative DELs.





## 2.1 Underlying assumptions

To follow the industrial standards, certain safety factors need to be considered for the design of the fatigue tests, which are omitted in this work for simplification.

All described procedures in this work follow certain simplifying assumptions, which are listed below. If any of these assumptions would be considered non-applicable the methods described in this work would need to be adjusted accordingly:

1. Validity of Timoshenko (1921) beam theory with small deformations, i.e. neglectable in-plane warping of the blade sections, neglectable Brazier-effect (Brazier, 1927). Otherwise the sectional stiffness components would become dependent on these deformations. (e.g. Brazier-effect reduces outer dimensions which in turn reduces the bending stiffness)

2. Only longitudinal strain is considered, i.e. shear, through-thickness and transverse strains are assumed negligible. See appendix A1 for more details.

3. Longitudinal strain is only affected by bending moments and axial force, i.e. influence of the torque or shear loads (e.g. via Bend-twist-coupling) are assumed negligible. See appendix A2 for more details.

4. Prismatic beam response is assumed, i.e. tapering or other longitudinal changes, e.g. ply-drops, do not affect the strains.

5. Stress and strain are assumed proportional

6. Material fatigue damage adheres to linear damage accumulation (Palmgren, 1924; Miner, 1945)

7. Material fatigue damage adheres to linear stress-life, i.e. The Basquin curve exponent (Basquin, 1910) is independent of load levels and cycle numbers.

## 2.2 Load simulation

First the DLCs which are to be considered are chosen e.g. from the standard IEC 61400-1:2019. And corresponding aero-elastic turbine simulations are performed resulting in set of time series $f(t)$ for load distributions along the blade length, i.e. sectional bending moments $M_x(t)$, $M_y(t)$ and longitudinal force $F_z(t)$. These loads are derived for a local reference coordinate system, where the x-y-plane of the section is perpendicular to the blades beam axis, which includes following the orientation of any pre-bend or sweep of the blade. Also the coordinate systems position and orientation, in which the loads are reported in, need

to follow the blade deformation during simulation. Otherwise the longitudinal z-axis would not be perpendicular to the cross section plane anymore and the subsequently used beam theory formulas would not be valid. Here, it is assumed that, in the undeformed state, the projection of this local coordinate systems x-axis to the blades root section is parallel to the global lead-lag direction of the blade for any section, and does not follow the blades twist angle. Following the twist angle or other orientations are also possible procedures. For the same sections along the blade, for which the loads are derived, the following

properties are computed (see Fig. 2):





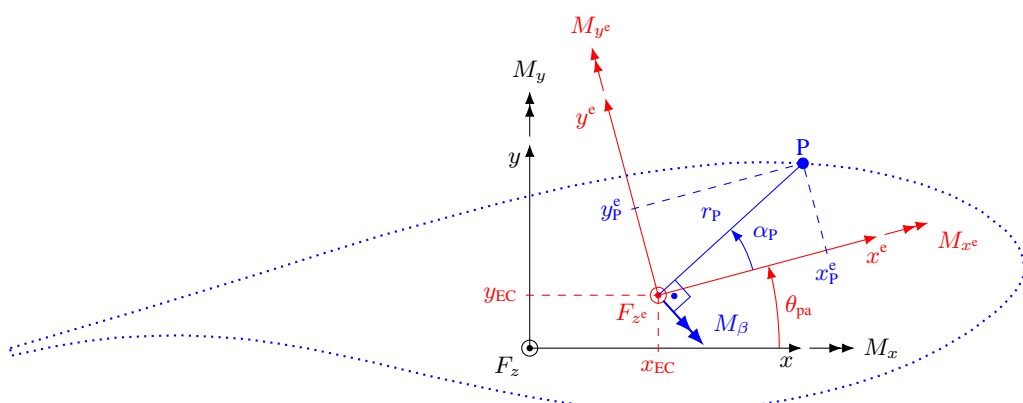

**Figure 2.** Relations between local reference coordinate system (black), elastic center coordinate system in principal orientation (red) and point P on the blade surface at angle $\alpha_P$ (blue) with corresponding variables and loads.

- Coordinates of the elastic center, i.e. the point where a force applied normal to the cross section will produce no bending curvatures: $x_{EC}$, $y_{EC}$

- Angle of principal stiffness axes orientation: $\theta_{pa}$ (also known as structural pitch)

- Principal bending stiffnesses about the $x^e$ and $y^e$ axis relative to elastic center: $EI_{x^e}, EI_{y^e}$

- Axial stiffness: $EA$

## 2.3 Transformation of load time series

The load time series are transformed to the elastic center and into the principal axes orientation of the corresponding section of the blade according to Eq. 1:

$$
\begin{bmatrix} M_{x^e} \\ M_{y^e} \\ F_{z^e} \end{bmatrix} = \begin{bmatrix} \cos\theta_{pa} & \sin\theta_{pa} & 0 \\ -\sin\theta_{pa} & \cos\theta_{pa} & 0 \\ 0 & 0 & 1 \end{bmatrix} \cdot \begin{bmatrix} 1 & 0 & -y_{EC} \\ 0 & 1 & x_{EC} \\ 0 & 0 & 1 \end{bmatrix} \cdot \begin{bmatrix} M_x \\ M_y \\ F_z \end{bmatrix} \tag{1}
$$

This load transformation is necessary as the following equations for strain are only valid for the elastic center in principal orientation (See Appendix A2).

From this the longitudinal strain at any given point of interest P within corresponding blade section (see Fig. 2) can be computed:

$$
\varepsilon_z(\mathrm{P}) = \frac{y_P^e}{EI_{x^e}} \cdot M_{x^e} - \frac{x_P^e}{EI_{y^e}} \cdot M_{y^e} + \frac{F_{z^e}}{EA} \tag{2}
$$





Assuming the longitudinal force contribution is negligible compared to the bending moment contribution, i.e. $\frac{F_z^{\mathrm{e}}}{EA} \approx 0$, and by
utilizing distance $r_{\mathrm{P}}$ from the elastic center to P and its angle $\alpha_{\mathrm{P}}$ the strain can be written as:

$$\varepsilon_{z,M}(\mathrm{P}) = \frac{r_{\mathrm{P}} \cdot \sin\alpha_{\mathrm{P}}}{EI_{x^{\mathrm{e}}}} \cdot M_{x^{\mathrm{e}}} - \frac{r_{\mathrm{P}} \cdot \cos\alpha_{\mathrm{P}}}{EI_{y^{\mathrm{e}}}} \cdot M_{y^{\mathrm{e}}} \tag{3}$$

In this work, the strain time series $\varepsilon_z(\mathrm{P}, t)$ are used as reference, as they are assumed to be the most realistic representation of
the materials fatigue behaviour.

The bending moment $M_\beta$ perpendicular to the direction of $r_{\mathrm{P}}$, which is usually assumed to contributing most to the strain
$\varepsilon_{z,M}$, can be calculated by coordinate transformation:

$$M_\beta = \sin\alpha_{\mathrm{P}} \cdot M_{x^{\mathrm{e}}} - \cos\alpha_{\mathrm{P}} \cdot M_{y^{\mathrm{e}}} \tag{4}$$

In the assumed conventional procedure only this bending moment $M_\beta$ is considered, especially only for the global blade main
directions, i.e. flapwise and lead-lag, which, under the assumed coordinate system orientation, correspond to $\alpha_{\mathrm{P},f} = -\theta_{\mathrm{pa}}$ and
$\alpha_{\mathrm{P},l} = -\theta_{\mathrm{pa}} + 90°$ respectively.

## 2.4 Rainflow counting

To accumulate the load time series from the simulation into corresponding DELs, the time series are converted via the Rainflow
counting algorithm (ASTM E1049-85) into a list of occurring load amplitudes $A_i$ with corresponding mean loads $M_i$ and cycle
numbers $n_i$. This list can be compressed further into so-called Markov-matrices, by sorting the loads into discrete intervals,
i.e. binning them. Here, no binning was applied.

## 2.5 Mean load correction

Only after Rainflow counting and before the next processing step the mean load correction (MLC) can be applied. This step
is necessary to account for the effect of the mean load on material fatigue. This entails changing a load amplitude $A_i$, which
corresponds to a specific mean load $M_i$, to a corrected load amplitude $A_{i,\mathrm{MLC}}$, which in turn corresponds to the mean load
$M_{i,\mathrm{MLC}} = 0$. This corrected amplitude is computed such that it contributes the same material fatigue damage as the original
$A_i$-$M_i$-pair. This correction requires the use of constant-life-diagrams (CLD) which are material specific.

The simplest form of this is a liner symmetric CLD (also known as Goodman- or Goodman-Haigh-diagram), which only
requires one ultimate load $U$ and assumes symmetric behaviour in tension and compression. For this the MLC can be performed
according to Eq. 5a:

$$A_{i,\mathrm{MLC}} = A_i \cdot \frac{U}{U - |M_i|} \tag{5a}$$

As most composite materials have different properties in tension and compression a shifted Goodman diagram is proposed
in DNV ST-0376:2024. This used different ultimate loads for tension $U_{\mathrm{t}}$ and compression $U_{\mathrm{c}}$, which results in Eq. 5b for the
MLC:

$$A_{i,\mathrm{MLC}} = A_i \cdot \frac{U_{\mathrm{avg}} - |U_{\mathrm{mid}}|}{U_{\mathrm{avg}} - |M_i - U_{\mathrm{mid}}|} \qquad \text{with} \quad U_{\mathrm{avg}} = \frac{|U_{\mathrm{t}} - U_{\mathrm{c}}|}{2}, \quad U_{\mathrm{mid}} = \frac{U_{\mathrm{t}} + U_{\mathrm{c}}}{2} \tag{5b}$$





Also more complex CLDs as proposed by Sutherland and Mandell (2005) can be employed. Since the required material properties are only available for stress or strain data, this correction is not possible for bending moments. Therefore, when employing the conventional bending moment based approach, the impact of the mean load cannot be taken into account and has to be neglected, resulting in Eq.5c:

$$A_{i,\mathrm{MLC}} = A_i \tag{5c}$$

**2.6  Linear damage accumulation**

After MLC the corrected load amplitudes $A_{i,\mathrm{MLC}}$ for each simulation are accumulated into a single DEL amplitude $A_{\mathrm{DEL}}$ with an arbitrary cycle number $N_{\mathrm{DEL}}$, using linear damage accumulation according to Palmgren (1924) and Miner (1945), assuming a linear stress-life according to Basquin (1910):

$$A_{\mathrm{DEL}} = \left( \frac{\sum\limits_i \left( n_i \cdot (A_{i,\mathrm{MLC}})^m \right)}{n_{\mathrm{DEL}}} \right)^{\frac{1}{m}} \tag{6}$$

where $m$ denotes the negative inverse Basquin curve exponent of the material under investigation.

There are several approaches how to define this arbitrary amount of cycles $n_{\mathrm{DEL}}$. Some research suggested to use the dominating frequency of the blade if it is contained in load spectrum or otherwise the zero/mean crossing frequency (Veers, 1982). Another approach was to pick up a frequency of 1Hz, which was representative for a turbine size (Madsen et al., 1984). The latter approach was widely adopted because simulation time in seconds is equal to the number of cycles and nowadays 1Hz

equivalent load is the commonly accepted practice, resulting in $n_{\mathrm{DEL}} = t$.

After the loads for each separate simulation $j$ are accumulated into one damage equivalent load $A_{\mathrm{DEL},j}$ each with $n_{\mathrm{DEL},j} = 1$ according to Eq. 6, the loads from different simulations are accumulated into one total damage equivalent load amplitude $A_{\mathrm{DEL,total}}$ with a cycle number of $N_{\mathrm{DEL,total}}$ using probabilities of occurrence as weighting factors:

$$A_{\mathrm{DEL,total}} = \left( \frac{\sum\limits_j \left( \frac{LT}{t_j} \cdot n_{\mathrm{DEL},j} \cdot p_{ws,j} \cdot p_{yaw,j} \cdot p_{\mathrm{DLC},j} \cdot (A_{\mathrm{DEL},j})^m \right)}{N_{\mathrm{DEL,total}} \cdot \sum\limits_j \left( n_{ts,j} \cdot p_{ws,j} \cdot p_{yaw,j} \cdot p_{\mathrm{DLC},j} \right)} \right)^{\frac{1}{m}} \tag{7}$$

where $LT$ denotes the total expected turbine design lifetime, $t_j$ the duration of the time series, $n_{ts,j}$ the number of turbulence seeds (i.e. the number of simulations with the same conditions), and $p_{ws,j}$, $p_{ws,j}$, $p_{\mathrm{DLC},j}$ the probabilities of the simulations wind speed, yaw angles, and design load case (DLC), respectively. If further variables are differentiated with more simulations the probabilities need to be adapted accordingly. As each simulation contains three blades, the loads from each blade can be considered as separate simulation runs. This effectively triples the number of turbulence seeds $n_{ts,j}$, if the loads for all three

blades are evaluated and accumulated.

Note, this damage accumulation is only valid for linear stress-life (assumption 7 in section 2.1). To consider more complex fatigue behaviour (e.g. Stüssi (1955); Rosemeier and Antoniou (2022)), the damage accumulation (Eq. 6 and 7) needs to be adjusted accordingly.





The resulting load DELs can then be used as target loads for blade fatigue testing. Dependent on the scope of the fatigue test, the amount of investigated angles $\alpha_\mathrm{P}$ and blade sections, has to be chosen correspondingly. The fatigue tests then have to be designed to match or exceed these target loads.

## 2.7  Methods for the enhanced procedure

From Eq. 3 and 4 it can be seen, that the strain and the swept bending moment are generally not proportional $\varepsilon_{z,M} \not\propto M_\beta$. There are only two cases when they are proportional: (I) When the two principal stiffnesses of the section are equal $EI_{x^\mathrm{e}} = EI_{y^\mathrm{e}}$, which is usually only the case at the cylindrical root of the rotor blade, or (case 2.1) When the position of interest P is on the principal axes, i.e. $\alpha_\mathrm{P} = 0°; \pm 90°; 180°$. As the conventional target loads are usually based on these bending moments and material fatigue damage is based on stresses or strains, this non-proportionality leads to the discrepancies between the fatigue loads in blade fatigue testing and material fatigue damage, which arise from the design loads.

Further discrepancies can arise if the moments are converted into DELs while omitting the load transformation into the elastic center (see path 0 in Fig. 1). The impact of this is outside the scope of this work as it is highly dependent on the arbitrary position of the used coordinate systems.

To mitigate the problem of non-proportionality of bending moments and strain Castro et al. (2021b, 2024) proposed the modified bending moment $M_\beta'$ to be used as basis for the target loads instead of the regular bending moments $M_\beta$ (see path 2.1 in Fig. 1). In this work, $M_\beta'$ has been slightly modified compared to Castro et al. (2021b) to closer represent strain values:

$$M_\beta' = \sin \alpha_\mathrm{P} \cdot M_{x^\mathrm{e}} - \cos \alpha_\mathrm{P} \cdot \frac{EI_{x^\mathrm{e}}}{EI_{y^\mathrm{e}}} \cdot M_{y^\mathrm{e}} \tag{8}$$

Translating the loads into $M_\beta'$ for the test design instead of transforming the data directly into strains has the benefit, that the data required for the translation does not contain sensitive design data of the blade, which helps with data transfer between OEM and test center, as highlighted by Castro et al. (2021b). With information on geometry and stiffness properties $M_\beta'$ can be transformed into the corresponding strain:

$$\varepsilon_{z,M} = \frac{r_\mathrm{P}}{EI_{x^\mathrm{e}}} \cdot M_\beta' \tag{9}$$

This transformation is only valid if the assumption of the contribution of the longitudinal force to the strain being negligible holds. Therefore the impact of this assumption is investigated in section 3.1.

For the MLC of $M_\beta'$, Castro et al. (2021b) proposed an approach based on the symmetric Goodman-Haigh diagram, but without the use of material data but rather unspecified ultimate loads derived from experience of the test institution. Also, the symmetry which only requires angles $\alpha_\mathrm{P} = 0°...180°$ does not hold anymore, once the MLC is applied and angles $\alpha_\mathrm{P} = -180°...180°$ are required. As Eq. 9 allows for simple conversion between strain $\varepsilon_{z,M}$ and $M_\beta'$, this enables the use of MLC in the $M_\beta'$-domain with material based data by converting the CLD-data into the $M_\beta'$-domain (see path 2.2 in Fig. 1). For example, to enable Eq. 5b, the ultimate tension and compression loads need to be evaluated:

$$M_{\beta,Ut}' = \frac{EI_{x^\mathrm{e}}}{r_\mathrm{P}} \cdot \varepsilon_{z,Ut}, \qquad M_{\beta,Uc}' = \frac{EI_{x^\mathrm{e}}}{r_\mathrm{P}} \cdot \varepsilon_{z,Uc} \tag{10}$$





Though this requires the derivation of individual CLD-data for every position of interest along the blade. Also $M'_\beta$ after MLC is not independent of $r_P$ anymore and is only valid for the position for which the corresponding CLD-data were derived. If multiple positions along the $\alpha_P$-direction with different $r_P$ are of interest, multiple CLDs have to be considered for the same $M'_\beta$. But the benefit of confidentiality still holds, as no direct material data need to be known for this MLC.

## 3   Case studies - Investigation of assumptions and methods

To demonstrate the difference between the methods described above, the 138m long reference blades of the IEA 22MW offshore reference wind turbine (Zahle et al., 2024) are used here as a test case. Load time series are generated from aero-elastic simulations using HAWC2 (Larsen and Hansen, 2024) of different design load cases of this turbine. These simulations cover wind speeds from 3-25m/s in 1m/s steps, with yaw-misalignment of 0°, 8° and 352°, and six turbulence seeds each, while considering all three blades ($n_{ts} = 18$). These simulations represent the power production design load case with normal

turbulence model (DLC 1.2) according to IEC 61400-1:2019. For the design and certification further load cases of the turbine concerning power-loss during production (DLC 2.4), start-up (DLC 3.1), shut-down (DLC 4.1) and parked (DLC 6.4) need to be considered. However, the standard doesn't specify individual contributions of these load cases in the turbine lifetime and leave this decision to the designer. All service and emergency load cases are design dependent, whereas DLC 1.2 mainly depends on probability of wind speed distribution between cut-in and cut-out speeds and is considered to occur approximately

95% of the turbine lifetime (Gözcü and Verelst, 2020). Therefore, for simplification, the other DLCs are not considered in this study. This results in a total number of 1242 load time series of $t = 600$s each with a resolution of 0.01s. These load-time series are evaluated at 49 sections along the blade span, for which the cross sectional stiffness properties are known. These time series are evaluated as described above.

For simplification of the test case the material of the blade is assumed to be uniax carbon fibre reinforced polymer (CFRP) on

the spar caps and uniax glass fibre reinforced polymer (GFRP) everywhere else in the blade. The assumed material parameters for the evaluation are listed in Table 1. The MLC in this study is performed utilizing Eq. 5b.

**Table 1.** Fatigue properties of materials (IEC 61400-5:2020; Zahle et al., 2024; Camarena et al., 2022).

| Material | $m$ | $\varepsilon_{z,Ut}$ | $\varepsilon_{z,Uc}$ |
|----------|-----|------|------|
| CFRP | 14 | 0.0160 | -0.0110 |
| GFRP | 10 | 0.0255 | -0.0148 |

In this example the local reference coordinate system of the sections have the same orientation as the global blade coordinate system only following the blades pre-bend and deformation. The sweep angle $\varphi$ in the following result plots is defined as $\varphi = \alpha_P + \theta_{pa}$ measured from the elastic center.





Here, the loads were evaluated as described above for all sweep angles $\varphi$ between -180° and 180° in 0.5° steps. For each angle highest $r_P$, i.e. the most outer shape of the blade was used as this has the highest strain and is assumed to be the most critical.

The loads for each separate simulation $j$ are accumulated individually according to Eq. 6 with $n_{\mathrm{DEL,j}} = 1$. As only the DLC1.2 is used in this study, Eq. 7 is adjusted here as shown in Eq. 11, to combine the different simulation results into one damage equivalent load amplitudes

$$A_{\mathrm{DEL,total}} = \left( \frac{\sum_j \left( \frac{LT}{t_j} \cdot p_{ws,j} \cdot p_{yaw,j} \cdot (A_{\mathrm{DEL},j})^m \right)}{N_{\mathrm{DEL,total}} \cdot 0.95 \cdot \sum_j \left( n_{ts,j} \cdot p_{ws,j} \cdot p_{yaw,j} \right)} \right)^{\frac{1}{m}} \tag{11}$$

with a turbine lifetime of $LT = 20$ years and $N_{\mathrm{DEL,total}} = 2$ million. For the wind speeds probabilities $p_{ws}$ the Weibull-distribution with a shape parameter of 2 and a scale parameter of 11.28. For the yaw angle probabilities of $p_{yaw}(0°) = 0.5$, $p_{yaw}(8°) = 0.25$ and $p_{yaw}(352°) = 0.25$ are used. In the following the impact on the results of a few specific optional components of the evaluation procedure are investigated.

### 3.1 Impact of longitudinal force contribution

The first study investigates the assumption of negligibility of the longitudinal force. Therefore the resulting strain DELs including MLC without the longitudinal force strain $\varepsilon_{z,M,\mathrm{DEL,MLC}}$ contribution and with it $\varepsilon_{z,\mathrm{DEL,MLC}}$ are compared. These measures are evaluated in the exact same way as shown Fig. 1 (path 3.2) with the only difference of utilizing Eq. 2 or Eq. 3, respectively. The relative difference between them are shown in Fig. 3, where they are projected on the blade geometry (left) and plotted over blade span and angle with the trailing edge (TE), the leading edge (LE) and the boundaries of the spar caps on suction side (SS-SC) and on pressure side (PS-SC) are marked for reference. The results show that considering the influence of the longitudinal force, compared to neglecting it, raises the accumulated DELs on the suction side by max. 1.8% and lowers it by min. -1.8% on the pressure side especially close to the root. This deviation is considered negligible and confirms the assumption that the longitudinal force can be neglected.

### 3.2 Impact of mean load correction

The next study investigates the impact of the MLC on the accumulated DELs. Therefore the DELs are evaluated with and without MLC, utilizing Eq. 5c and Eq. 5b (path 3.1 and 3.2 in Fig. 1), respectively. The relative difference between the modified bending moment DELs are shown in Fig. 4. Due to their proportionality the same differences are found for the strain, i.e. $\frac{\varepsilon_{z,M,\mathrm{DEL,MLC}}}{\varepsilon_{z,M,\mathrm{DEL}}} = \frac{M'_{\beta,\mathrm{DEL,MLC}}}{M'_{\beta,\mathrm{DEL}}}$. The results show that the DELs along the LE and TE are not affected by the MLC. However, on the SS-panels, the MLC raises the DELs by up to 7.5%, and on the whole SS-SC between 20m and 100m it ranges from about 8% to up to 10.4% around the 80m span. On the PS, up to 110m span, the DELs are lowered by at least -3%, with the lowest of -6.1% on the spar cap around 25m span. These deviation are considered significant, and the increased load especially confirms the necessity of MLC. Using the conventional methods without MLC to define the target loads would therefore lead



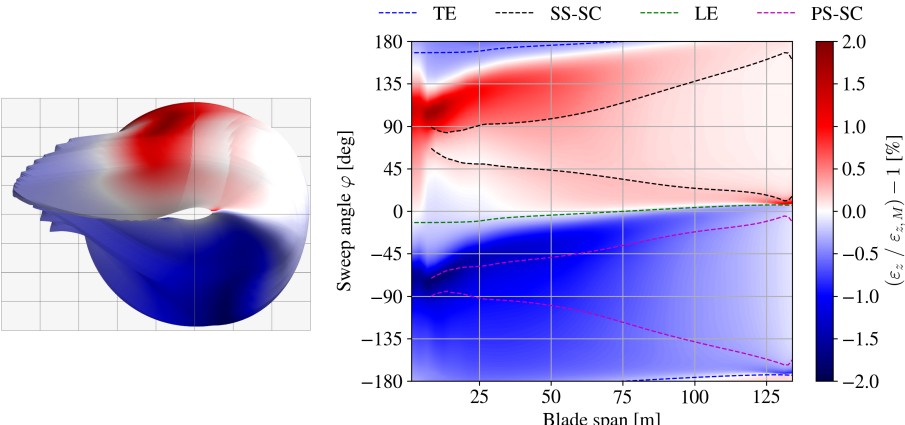

**Figure 3.** Distribution of relative difference between damage equivalent accumulated longitudinal strain amplitude (incl. MLC) with ($\varepsilon_{z,\text{DEL,MLC}}$) and without ($\varepsilon_{z,M,\text{DEL,MLC}}$) the consideration of longitudinal force contribution.

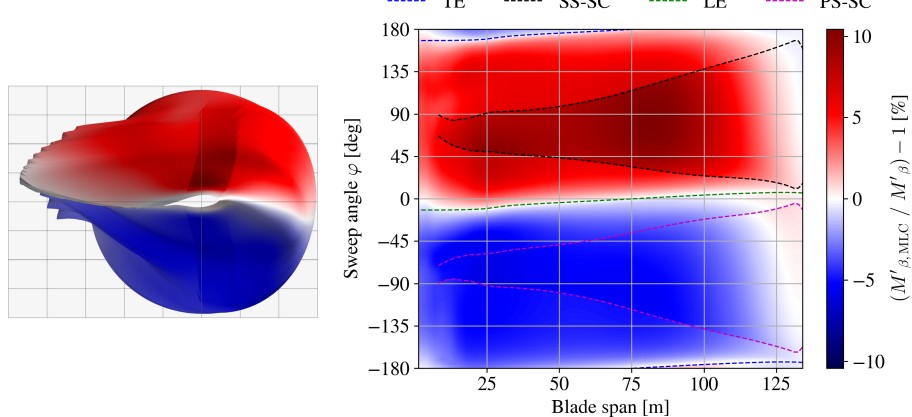

**Figure 4.** Distribution of relative difference between damage equivalent accumulated modified bending moment amplitude with ($M'_{\beta,\text{DEL,MLC}}$) and without ($M'_{\beta,\text{DEL}}$) mean load correction.

to an insufficiently tested SS-SC: A 9.4%-decrease (opposite to a 10.4%increase) of DELs leads to a 75% decrease of applied fatigue damage (with $m = 14$), which is missing compared to the predicted fatigue damage from the timeseries with MLC. This would lead to a fatigue test confirming only 25% of the intended design fatigue life. The 10.4% discrepancy shown in this study may also be exceeded when using different material properties or different CLD-formulations.

### 3.3 Impact of modified bending moment $M'_\beta$

To investigate the impact of using the modified bending moments $M'_\beta$ instead of the regular bending moments $M_\beta$ for defining target loads for fatigue testing, these measures cannot simply be compared by values, as their formulations are inherently



different. Hence, for comparison multiple sets of simplified uniaxial fatigue test loads for flapwise and lead-lag are computed here. These are designed to satisfy the different target loads respectively, i.e. match or exceed the corresponding field loads. As these are uniaxial tests only the loads in the global main directions, i.e. $\varphi = 180°, 0°, 90°, -90°$, are considered as targets.

In conventional fatigue test designs the blade is fixed at the root with the blade axis horizontally and with suction side facing downwards. Therefore, the blades self weight generates tension in the pressure side and compression in the suction side, which is assumed as mean load for the simplified test loads (neglecting any additional masses used in real fatigue tests, e.g. load frames). The cycle number for both tests is set to $n_{\text{test}} = 2e6$. As test amplitude the scaled load vector resulting from the first two natural bending mode shapes of the blade are considered (assuming $F_z = 0$), as fatigue tests are usually excited in

resonance:

$$
\boldsymbol{L}_M = \begin{bmatrix} M_x \\ M_y = 0 \\ F_z = 0 \end{bmatrix}_{\text{self weight}} \qquad \boldsymbol{L}_{A,f} = S_f \cdot \begin{bmatrix} M_x \\ M_y \\ F_z = 0 \end{bmatrix}_{\text{1st mode shape}} \qquad \boldsymbol{L}_{A,l} = S_l \cdot \begin{bmatrix} M_x \\ M_y \\ F_z = 0 \end{bmatrix}_{\text{2nd mode shape}} \tag{12}
$$

where $\boldsymbol{L}_M$ is the mean load vector, and $\boldsymbol{L}_{A,f}$ and $\boldsymbol{L}_{A,l}$ are the amplitude load vectors for flapwise and lead-lag test, respectively, with the corresponding scaling factors $S_f$ and $S_l$. The amplitude load vectors are scaled for both tests in such a matter, that the accumulated load from both tests satisfy the target loads. Therefore the test loads are evaluated in the same way as the field

loads starting from Eq. 1. This is done for each section along the blade individually and independent from each other. To find the right scaling factors for each section an optimization problem needs to be solved:

$$
\underset{S_f, S_l}{\text{minimize}} \quad \sum_{\varphi \in \Phi} \left( A_{\text{DEL},\varphi,\text{test}}(\boldsymbol{L}_M, \boldsymbol{L}_{A,f}, \boldsymbol{L}_{A,l}) - A_{\text{DEL},\varphi,\text{field load}} \right) \tag{13a}
$$

$$
\text{subject to} \quad A_{\text{DEL},\varphi,\text{test}}(\boldsymbol{L}_M, \boldsymbol{L}_{A,f}, \boldsymbol{L}_{A,l}) \geq A_{\text{DEL},\varphi,\text{field load}}, \quad \varphi \in \Phi = \{180°, 0°, 90°, -90°\} \tag{13b}
$$

This results in load amplitude distribution for each test. Note, that these load distributions do not represent actual fatigue

tests, which could be performed in reality, but rather the best case scenario where the target loads are matched as close as possible along the whole blade span. This optimization is executed to generate test loads designed for three different cases (case 1, 2.1 and 2.2 corresponding to paths in Fig. 1) to match the corresponding field load data for $A_{\text{DEL}} = M_{\beta,\text{DEL}}$ (case 1), $M'_{\beta,\text{DEL}}$ (case 2.1) or $M'_{\beta,\text{DEL,MLC}}$ (case 2.2).

From this, the impact of using these different approaches can be evaluated by evaluation the test loads in terms of damage

equivalent strain amplitude $\varepsilon_{z,\text{DEL,MLC}}$. Fig. 5 shows the difference of the test loads for case 2.1 and 2.2, each relative to case 1. Designing the test loads for $M'_{\beta,\text{DEL}}$ (case 2.1) compared to $M_{\beta,\text{DEL}}$ (case 1) requires generally required higher loads to achieve the target. In lead-lag direction (i.e. the load needs to be raised by up to 14% around 25m blade length, which corresponds to the maximum chord length. Towards the tip outboard of 107m the load needs to be raised even more, though this area is usually not within the area of interest of fatigue testing. Only in the area between 77m and 96m the load needs to be lowered. In flapwise

direction the load only needs to be raised by up to 3% close to the root around 15m. Between 60m and 90m the flapwise loads for case 1 and 2.1 are almost identical. Designing the loads for $M'_{\beta,\text{DEL,MLC}}$ (case 2.2), i.e. considering MLC, requires even





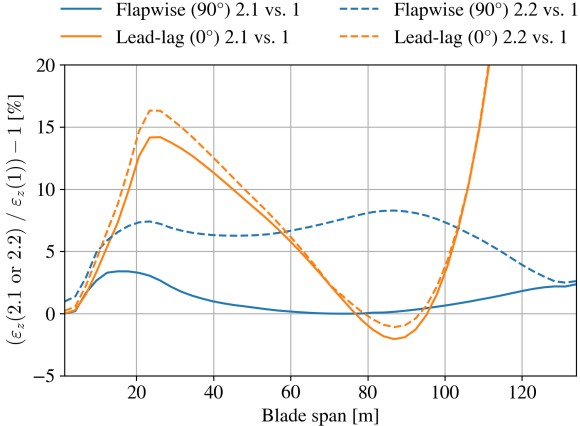

**Figure 5.** Distribution of relative difference between damage equivalent accumulated longitudinal strain amplitude (incl. MLC) $\varepsilon_{z,M,\mathrm{DEL,MLC}}$ for test loads designed for case 1, 2.1 and 2.2.

higher loads. In the lead-lag direction the loads are similar to case 2.1 and need to be raise by up to 16% around 25m compared to case 1. In flapwise direction, the load is raised by 5% to 8% almost along the whole blade (8m-115m) compared to case 1.

This shows the assumed conventional method to design fatigue test loads (case 1) can lead to severe under-testing of the
blade, as the more detailed methods (case 2.1 and 2.2) require up to 16% higher loads.

To elaborate further on these differences, Fig. 6 shows the the fatigue damage from case 1 relative to the damage from case 2.1 and 2.2 respectively. This damage ratio is derived from the load ratio and $m$ for the corresponding material: $\frac{D(1)}{D(2.1\ \mathrm{or}\ 2.2)} = \left(\frac{\varepsilon_{z,\mathrm{DEL,MLC}}(1)}{\varepsilon_{z,\mathrm{DEL,MLC}}(2.1\ \mathrm{or}\ 2.2)}\right)^{m}$. This reveals, that the 16% required load raise in lead-lag direction corresponds to a fatigue damage deficit of just under 80%. In the flapwise direction between 15m and 105m case 1 only deals 33%-43% of the fatigue damage
of case 2.2.

Though, this method only considered the main flapwise and lead-lag blade directions, i.e. $\varphi = 180°, 0°, 90°, -90°$. If other directions are also considered the test loads compared to field loads for case 1 and 2.2 are shown in Fig. 7 and Fig. 8 , respectively. For case 1 the loads along the main directions are not matched as suggested above, but for case 2.2 the loads along the main directions are tested sufficiently. But both test scenarios show large areas, which are loaded less from the test than
from the field loads (hatched areas). This suggests uniaxial fatigue testing is not sufficient to test the whole blade and more sophisticated testing methods, e.g. biaxial testing, will be required to test the whole blade sufficiently.

## 4 Implementation of target loads in testing

Any of the described accumulated loads (bottom row in Fig. 1) can be used as target loads for fatigue testing. Therefore, the test loads need to be transformed and evaluated in the same manner as the chosen target loads. These evaluated test loads can
then be checked if they meet or exceed the corresponding target loads within the areas of interest along the blade. This load



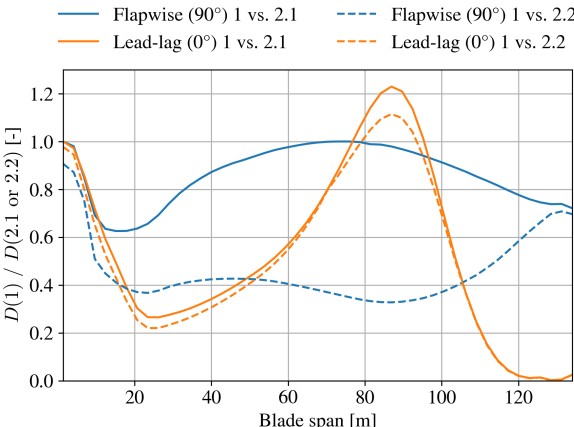

**Figure 6.** Distribution of fatigue damage ratio between case 1 and case 2.1 or case 2.2 respectively.

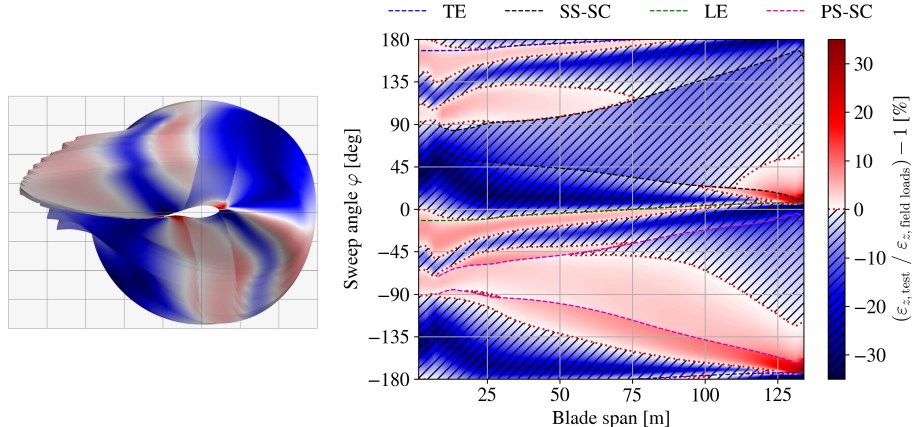

**Figure 7.** Distribution of relative difference between damage equivalent accumulated longitudinal strain amplitude (incl. MLC) $\varepsilon_{z,M,\text{DEL,MLC}}$ for conventional approach test loads (case 1) and field loads.

evaluation needs to be done during the fatigue test execution and also within the test design to be able to compare against the targets.

The conventional bending moment based approach (Fig. 1 path 0) for uniaxial fatigue test execution does not require a lot of processing as only the constant bending moment amplitude in the main directions with the corresponding cycle number needs to be evaluated. But as shown above, the error of this method can be magnificent. To adopt the proposed approach of this work (Fig. 1 path 2.2) for a test method with constant amplitude the bending moments measured during testing need to be transformed and processed according to Eq. 1, 8 and 5 same as the target loads with accumulation (Eq. 6) of the different sequential tests, e.g. flapwise and lead-lag test. Live Rainflow-counting with the corresponding accumulation of the test loads



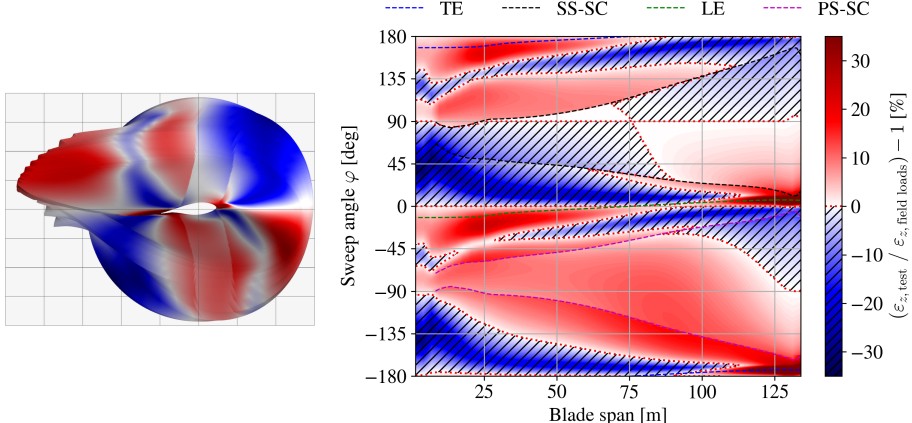

**Figure 8.** Distribution of relative difference between damage equivalent accumulated longitudinal strain amplitude (incl. MLC) $\varepsilon_{z,M,\mathrm{DEL,MLC}}$ for enhanced approach test loads (case 2.2) and field loads.

is only required for testing methods involving constantly changing load amplitudes, such as biaxial testing with an arbitrary
frequency ratio.

Using strains $\varepsilon_z$ including MLC (Fig. 1 path 3.2) as target loads will lead to the same results as the proposed approach, but it requires more potentially confidential data. In order for the testing facility to design and evaluate the test detailed geometric data and strain based CLDs would need to be shared by the OEM. Castro et al. (2024) showed, that by using the modified bending moment $M'_\beta$ less confidential data is required. Using the proposed modified bending moment $M'_\beta$ including MLC
(Fig. 1 path 2.2) requires different transformed CLDs for each target load which can be provided solely for the expected load levels of the corresponding $M'_\beta$ and are therefore more anonymised than the strain based data.

## 5   Conclusions

This work demonstrated that conventional methods for separated flapwise and lead-lag fatigue test sequences and target load evaluation can lead to substantial under-testing across major areas of rotor blades, potentially compromising the intended design
validation. It is proposed to employ the modified bending moment $M'_\beta$ as described by Castro et al. (2021b), in combination with a suitable mean load correction (MLC), to achieve a realistic load representation. This enhanced approach can be utilized to define necessary test loads that result in sufficient fatigue damage throughout all of the blade. The findings suggest that applying this methodology can require increases in target loads of up to 16% for uniaxial testing compared to the conventional methods. Future research should focus on practical implementations of the proposed methodologies into standardized testing
practices and explore additional testing techniques, such as biaxial testing, to apply the required loads on the whole blade and to further enhance the understanding of blade fatigue behavior.



## Appendix A:  Load assumption details

### A1    Relevant strain tensor components

The local strain tensor $\boldsymbol{\varepsilon}(s,t,n)$ (spanwise, transverse, normal) at any position $(s,t,n) \rightarrow (x,y,z)$ in a material consists of

six components. Considering these components based on a rotor blade section these are the spanwise strain $\varepsilon_s$, the through-thickness and transverse normal strains $\varepsilon_n$ and $\varepsilon_t$, and the in- and out-of-plane shear strains $\varepsilon_{tn}$ and $\varepsilon_{st}$, $\varepsilon_{sn}$.

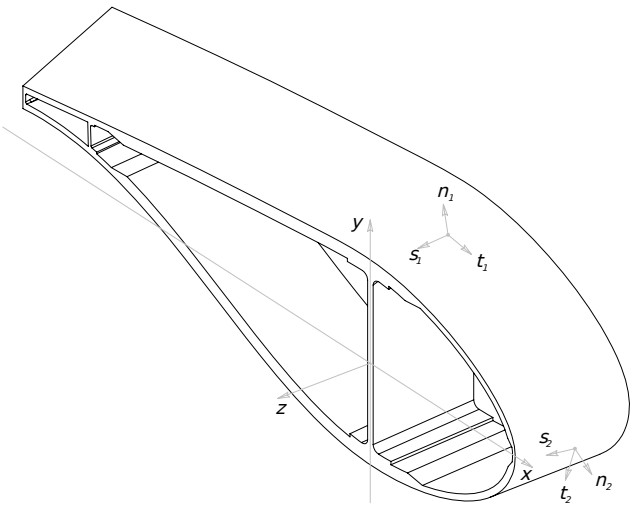

**Figure A1.** Coordinate systems for strain analysis of a blade section

In reality only $\varepsilon_s$, $\varepsilon_t$ and $\varepsilon_{st}$ can be directly measured by strain gauges on outer and inner blade surfaces. For tapered regions, the strain tensor should be transformed but there are not enough measured components. Therefore with composite anisotropy in mind and assuming that blades tapering is modest in practice $\varepsilon_z = \varepsilon_s$. This local longitudinal strain $\varepsilon_z$ is mainly influenced

by the bending curvatures and the longitudinal strain of a blade section (See A2 below). It is the main object of interest in this work.

The transverse and through-thickness $\varepsilon_t, \varepsilon_n$ strains result from cross sectional in plane deformation, which usually comes from Poisson effect, Brazier effect or local instability (e.g. panel buckling) or local bending. Transverse panel bending is the main contributor for $\varepsilon_t$ and reaction forces in the shear webs are the main contributor for $\varepsilon_n$ in the outer shell. The latter should

happen only at very high longitudinal strain levels, therefore it is only relevant for ultimate load cases but not for fatigue. The in-plane shear strain $\varepsilon_{tn}$ can only be caused by in-plane warping deformations and usually can be found in the cross sectional geometry with open cells. As these in-plane deformations are assumed to be negligible in beam theory (Assumption 1 in Section 2.1) the transverse and through-thickness strains also have to be assumed negligible $\varepsilon_t = 0$, $\varepsilon_n = 0$, $\varepsilon_{tn} = 0$.

The out-of-plane shear strains $\varepsilon_{sn}$ and $\varepsilon_{st}$ are caused by transverse shear forces and torsion of the blade. These cannot

generally be assumed to be zero, but they are not considered in this study for simplification, because their recovery from load signals is complex and only limited information on shear and multiaxial fatigue of composites is available.





## A2 Strain derivation for fully populated beam element

The symmetric 6x6 stiffness matrix of a beam cross section denoted $\boldsymbol{K}$ couples the cross sectional load and deformation vectors $\boldsymbol{L}$ and $\boldsymbol{\xi}$ (see Eq. A1). It contains the stiffness terms and can be fully populated for a composite beam. The coupling

(non-diagonal) terms can come from the geometry and the layup of the beam structure. In practice several of the coupling's terms are zero or very small. This 6x6 matrix is applied to generate the 12x12 beam element utilized in aero-elastic code, such as HAWC2 (Larsen and Hansen, 2024).

For the 6x6 cross-sectional stiffness matrix of rotor blades it is usually assumed that there are no couplings between longitudinal strain and shear forces, no couplings between bending curvatures and shear forces and no coupling between longitudinal

strain and the torsional moment. From classical laminated plate theory (CLPT) according to Reddy (2003), it is known that a laminate stacking sequence which is balanced, unidirectional or consist of cross-plies has no extension/shear couplings ($A_{16}$ and $A_{26}$ become zero in the in extensional stiffness matrix denoted $\boldsymbol{A}$). A laminate with extension/shear couplings will distort in the curing process, which is not desirable.

The bending-extension coupling stiffness matrix denoted $\boldsymbol{B}$, contains the coupling between bending/twisting curvatures and

extension/shear loads. In symmetric laminates all terms in the $\boldsymbol{B}$ matrix are zero. Symmetric laminates are almost always used as these effects are usually undesirable, as also laminates with a non-zero $\boldsymbol{B}$ matrix will distort in the curing process due to internal stresses, compromising the geometry of the blade surface (Jones, 2018). Furthermore, when fiber mats or plies are placed at an angle it results in reduced stiffness and reduced load carrying capacity towards the dominating longitudinal loading/strain. Therefore, wind turbine blades are typically designed without placing fiber mats or plies in an off-axis

orientation, however when placing the fibers mats in the mold during manufacturing smaller angles can arise from draping effects, mainly where the mold has double curvatures (normally in the region of maximum chord length). With the considerations above concerning $\boldsymbol{A}$ and $\boldsymbol{B}$ the corresponding coupling terms in the stiffness matrix $\boldsymbol{K}$ can be assumed to be zero: $K_{13} = 0$, $K_{14} = 0$, $K_{15} = 0$, $K_{23} = 0$, $K_{24} = 0$, $K_{25} = 0$, $K_{36} = 0$.

Furthermore, bend-twist-couplings are also assumed to be negligible. From CLPT it is known that these result from a part

of the laminate bending stiffness matrix denoted $\boldsymbol{D}$ ($D_{16}$ and $D_{26}$), if plies are oriented off-axis (in this case, off-axis with respect to the blade axis) (Jones, 2018). Assuming their negligibility leads to: $K_{46} = 0$, $K_{56} = 0$.

Combining the above assumptions leads to a reduced stiffness matrix $\boldsymbol{K}$:

$$\boldsymbol{L} = \boldsymbol{K} \cdot \boldsymbol{\xi} = \begin{bmatrix} F_x \\ F_y \\ F_z \\ M_x \\ M_y \\ M_z \end{bmatrix} = \begin{bmatrix} K_{11} & K_{12} & 0 & 0 & 0 & K_{16} \\ K_{12} & K_{22} & 0 & 0 & 0 & K_{26} \\ 0 & 0 & K_{33} & K_{34} & K_{35} & 0 \\ 0 & 0 & K_{34} & K_{44} & K_{45} & 0 \\ 0 & 0 & K_{35} & K_{45} & K_{55} & 0 \\ K_{16} & K_{26} & 0 & 0 & 0 & K_{66} \end{bmatrix} \cdot \begin{bmatrix} \gamma_x \\ \gamma_y \\ \varepsilon_z \\ \kappa_x \\ \kappa_y \\ \kappa_z \end{bmatrix} \tag{A1}$$



Translating both the load vector $\boldsymbol{L}$ and deformation vectors $\boldsymbol{\xi}$ to the cross-sectional elastic center as reference point and

rotate them to the principle bending axis orientation, eliminates further coupling terms $K_{34}^{\mathrm{e}} = 0$, $K_{35}^{\mathrm{e}} = 0$, $K_{45}^{\mathrm{e}} = 0$:

$$
\boldsymbol{L}^{\mathrm{e}} = \boldsymbol{K}^{\mathrm{e}} \cdot \boldsymbol{\xi}^{\mathrm{e}} =
\begin{bmatrix} F_{x^{\mathrm{e}}} \\ F_{y^{\mathrm{e}}} \\ F_z \\ M_{x^{\mathrm{e}}} \\ M_{y^{\mathrm{e}}} \\ M_{z^{\mathrm{e}}} \end{bmatrix} =
\begin{bmatrix}
K_{11}^{\mathrm{e}} & K_{12}^{\mathrm{e}} & 0 & 0 & 0 & K_{16}^{\mathrm{e}} \\
K_{12}^{\mathrm{e}} & K_{22}^{\mathrm{e}} & 0 & 0 & 0 & K_{26}^{\mathrm{e}} \\
0 & 0 & K_{33} & 0 & 0 & 0 \\
0 & 0 & 0 & K_{44}^{\mathrm{e}} & 0 & 0 \\
0 & 0 & 0 & 0 & K_{55}^{\mathrm{e}} & 0 \\
K_{16}^{\mathrm{e}} & K_{26}^{\mathrm{e}} & 0 & 0 & 0 & K_{66}^{\mathrm{e}}
\end{bmatrix} \cdot
\begin{bmatrix} \gamma_{x^{\mathrm{e}}} \\ \gamma_{y^{\mathrm{e}}} \\ \varepsilon_{z^{\mathrm{e}}} \\ \kappa_{x^{\mathrm{e}}} \\ \kappa_{y^{\mathrm{e}}} \\ \kappa_z \end{bmatrix}
\tag{A2}
$$

Inverting this reduced stiffness matrix $\boldsymbol{K}^{\mathrm{e}}$ leads to a reduced compliance matrix $\boldsymbol{C}^{\mathrm{e}}$:

$$
\boldsymbol{\xi} = (\boldsymbol{K}^{\mathrm{e}})^{-1} \cdot \boldsymbol{L}^{\mathrm{e}} = \boldsymbol{C}^{\mathrm{e}} \cdot \boldsymbol{L}^{\mathrm{e}} =
\begin{bmatrix} \gamma_{x^{\mathrm{e}}} \\ \gamma_{y^{\mathrm{e}}} \\ \varepsilon_{z^{\mathrm{e}}} \\ \kappa_{x^{\mathrm{e}}} \\ \kappa_{y^{\mathrm{e}}} \\ \kappa_z \end{bmatrix} =
\begin{bmatrix}
C_{11}^{\mathrm{e}} & C_{12}^{\mathrm{e}} & 0 & 0 & 0 & C_{16}^{\mathrm{e}} \\
C_{12}^{\mathrm{e}} & C_{22}^{\mathrm{e}} & 0 & 0 & 0 & C_{26}^{\mathrm{e}} \\
0 & 0 & \frac{1}{K_{33}} & 0 & 0 & 0 \\
0 & 0 & 0 & \frac{1}{K_{44}^{\mathrm{e}}} & 0 & 0 \\
0 & 0 & 0 & 0 & \frac{1}{K_{55}^{\mathrm{e}}} & 0 \\
C_{16}^{\mathrm{e}} & C_{26}^{\mathrm{e}} & 0 & 0 & 0 & C_{66}^{\mathrm{e}}
\end{bmatrix} \cdot
\begin{bmatrix} F_{x^{\mathrm{e}}} \\ F_{y^{\mathrm{e}}} \\ F_z \\ M_{x^{\mathrm{e}}} \\ M_{y^{\mathrm{e}}} \\ M_{z^{\mathrm{e}}} \end{bmatrix}
\tag{A3}
$$

From this, the longitudinal strain equation for all surface points P $(x_{\mathrm{P}}^{\mathrm{e}}, y_{\mathrm{P}}^{\mathrm{e}})$ can be derived as

$\varepsilon_{z,\mathrm{P}}(x_{\mathrm{P}}^{\mathrm{e}}, y_{\mathrm{P}}^{\mathrm{e}}) = \kappa_{x^{\mathrm{e}}} \cdot y_{\mathrm{P}}^{\mathrm{e}} - \kappa_{y^{\mathrm{e}}} \cdot x_{\mathrm{P}}^{\mathrm{e}} + \varepsilon_{z^{\mathrm{e}}} = \dfrac{M_{x^{\mathrm{e}}} \cdot y_{\mathrm{P}}^{\mathrm{e}}}{K_{44}^{\mathrm{e}}} - \dfrac{M_{y^{\mathrm{e}}} \cdot x_{\mathrm{P}}^{\mathrm{e}}}{K_{55}^{\mathrm{e}}} + \dfrac{F_z}{K_{33}}$
    $\tag{A4}$

where $K_{33} = EA$, $K_{44}^{\mathrm{e}} = EI_{x^{\mathrm{e}}}$, $K_{55}^{\mathrm{e}} = EI_{y^{\mathrm{e}}}$.

*Author contributions.* DM prepared the code and conducted the analysis. SS and PB supported in developing the methods and preparing the case study. KB supervised and guided in the conception of the methods. EP supervised DM and acquired funding. DM prepared the manuscript with contributions from all co-authors.

*Competing interests.* The authors declare that they have no conflict of interest.

*Acknowledgements.* We acknowledge the support by Frederick Zahle and Athanasios Barlas from DTU with the load simulation and generation of load timeseries for the reference turbine test case. We also acknowledge the support provided by the German Federal Ministry for Economic Affairs and Energy (BMWE) within the SmarTestBlade project (03EE2037)



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
