# Peer review of "Enhanced approach to match damage-equivalent loads in rotor blade fatigue testing"

_Wind Energy Science, 2025_

## Referee Comment (RC3)

Manuscript Number: WES-2025-99

Title: Enhanced approach to match damage-equivalent loads in rotor blade fatigue testing

General Comments: Overall, the manuscript well written. The introduction covers the background and literature review of relevant topic areas related to the research. The description, comparison and proposed methodology to define fatigue test loads are thoroughly discussed with sound conclusions. However, a few minor clarifications and corrections could enhance the manuscript.

Specific Comments:

| Comment | Line | Comment |
|---|---|---|
| 1 | 80 | Please include a discussion of who should be interested in this research. |
| 2 | 84 | Figure 1 identifies, and the text briefly describes the load time series processing paths. However, none of the nomenclature (i.e. abbreviations, variables, symbols, subscripts) in Figure 1 are described. To aid the reader, please define all terms used in Figure 1 in the Section 2 text or include a nomenclature section at the beginning of the manuscript. |
| 3 | 186 | The manuscript states " … and $p_{ws,j}, p_{ws,j}, p_{DLC,j}$ …" The second instance of $p_{ws,j}$ should be $p_{yaw,j}$. |
| 4 | 199-200 | The manuscript states "… only two cases when they are proportional: (I) When … or (case 2.1) When …". Please clarify if "case 2.1" refers to the processing path 2.1 or is it the second case of proportionality and therefore should be labeled as "(II)". |
| 5 | 242, 244 | The test case assumes the blade material to be only uniax CFRP and GFRP. However, the IEA 22MW Reference Wind Turbine (RWT) blade contains uniax CFRP and uniax, biax, and triax GFRP, as well as medium density foam. Please clarify how the correct blade cross sectional stiffness was maintained using only uniax CFRP and GFRP. Also, while the IEA 22MW RWT does not include adhesive joints, please comment on the lack of adhesive joints in the test case and proposed methodology to define fatigue test loads. |
| 6 | 283 | The manuscript states "… may also be exceeded when using different material properties or different CLD-formulations." This is a good statement and appears to partially address the concern in Comment 5 regarding simplification of material choices in the test case. |
| 7 | 301, 306 | Please identify and describe the optimization methodology executed. |
| 8 | 309 | The manuscript statement "… evaluated by evaluation the test loads …" is not grammatically correct. Please correct. Possible corrections are "… evaluated by evaluation of the test loads …", "… evaluated by evaluating the test loads …", or rewriting the statement to "… determined by evaluation the test loads …". |

| | | |
|---|---|---|
| 9 | 359 | The future research areas should also discuss the limitations of the proposed methodologies and future research needed to address issues resulting from the combination of research assumptions (in Section 2.1), material simplification (in Section 3) and assumptions of negligibility (in Appendices A1 and A2). |
| 10 | 393 | The manuscript states "… zero in the in extensional …". The second "in" is not required. |

---

## Author Comment (AC1)

**Response letter on comments regarding manuscript "Enhanced approach to match damage-equivalent loads in rotor blade fatigue testing", wes-2025-99**

The authors would like to thank the referees and the editor for their time, their valuable comments and the positive evaluation on the manuscript. The comments requiring changes are taken into account in the manuscript. The answers and the corresponding changes are listed below. Additionally, all applied changes are highlighted in the manuscripts below the answers. All line references in the answers refer to the highlighted manuscript.

Besides the changes proposed by the referee-comments, the english grammar and spelling of the manuscript was revised and some formulations were edited for better readability without changing the content. Most changes are shown in the diff below, only minor changes (e.g., comma-corrections) are not highlighted for clarity.

Please see our responses to the specific comments:

**Referee 1:**

Comment 1: Lines 70 - 80 there is a repetition of 'As the proposed approach can be used for any load direction, it enables target loads for any fatigue test method including biaxial testing. '

The sentence in line 80 was deleted.

Comment 3: Equations 2 and 3 needn't be used, there is a matrix formulation approach which can use an arbitrary reference line and the appropriate 'bending and axial'  $3 \times 3$  sub-matrix of the  $6 \times 6$  section matrix to calculate strain given loads defined at an arbitrary reference point/axis. If everything is done correctly, the two approaches will give the same result so the way described in the paper is valid but not the only way (having finished the review, I now see this is addressed very well in the appendices).

 We agree that this is covered in the appendix. We see no need to change the manuscript.

Comment 4: Line 250 - that's a lot of steps!

• This high angular resolution was chosen, to capture any discontinuities in the geometry. Lower resolutions may also be sufficient.

Comment 5: Line 265, section 3.1. This is an interesting comparison - I'm not sure our customers would agree that 1.8% is negligible, but with our damage-based approach (which from experience of discussing this with OEMs seems to be how it is done when confidentiality is not an issue, i.e. in-house) the test load (where we can't control Fz) is

just increased so that the damage (which is only really coming from Mx and My loading in the test) is matching the service life damage (coming from Mx, My and Fz loading).

• If strain targets are available the test can be based on those and Fz is considered. If only bending moments are available Fz cannot be included. Changes were made in the revised manuscript in line 301f and further information was added in lines 302-304.

Comment 6: This approach is good because it has the advantage of needing less customer data, and I suppose material properties for UD carbon, glass and biax and triax are available in the OPTIDAT database to get the UTS and UCS for all the common material types you'd see in the blade. The disadvantage is that you can't consider the 3 R-value diagram - when we do fatigue analyses we frequently see that carbon fails unless you can account for the very high values of m in the SN curve for compressive loading. Our test load derivation essentially continues on path 3.2 to iterate to find loads which do the same amount of damage for each cross section of blade.

• For the 3 R-Value diagram the mean load correction as described in Section 2.5 (i.e. Equation 5) need to be adapted accordingly, but the overall method does not require changes. This was mentioned already in Line 181 (165 in preprint). A sentence is added in line 181,182 to the manuscript to explain this more clearly.

Comment 10: Line 330- add text to draw attention to the criticality of the regions missed. Along the lines of "The hatched area includes features which should be tested such as critical structural details and significant load transitions between design elements."

A corresponding text was added in line 368f.

Comment 11: Line 340 -change "magnificent", could be changed to either significant or magnified.

• Magnificent" is changed to "significant" in line 381.

Comment 13: Line 353. Perhaps the conclusion could make reference to the different streams laid out in the flow diagram in fig 1.

References to Fig.1 are added to the conclusion in line 395-398.

Comment 14: Figure 1: "sweep" is used, as well as in line 114, I think these have 2 different meanings, but its not clear.

• To avoid this double meaning, the reference to the planform sweep of the blade tip was removed in line 127. The remaining "sweep" describes the angular sweep around the blade circumference (angle  $\alpha_P$ ).

Comment 15: Fig 3 and Fig 4 – do these relate to the streams labelled in the flow diagram? If so, they could be included in the caption.

• Yes, the comparisons partially follow different paths on the flow chart in Fig. 1. References to Fig. 1 were added to the captions of Fig. 3 and Fig. 4. Also a refence was added in line 309.

**Referee 2:**

Comment 2: Line 258, The yaw angle probabilities may represent deviations from the theoretical optimum under power production. Perhaps an explanation is needed for the choice of these specific probabilities.

Overall, there are no definitions in IEC 61400-1 standard available specifying these values and the designers must choose the values by their experience and related to the turbine operational conditions. Here, the yaw angle probabilities were chosen based on experience by DTUs aeroelastic group. These numbers are used for IEC reference turbine designs. For more on the topic of yaw misalignment follow next references e.g., <a href="https://doi.org/10.1002/we.1612">https://doi.org/10.1002/we.1612</a> and <a href="https://doi.org/10.1002/we.1739">https://doi.org/10.1002/we.1612</a> and <a href="https://doi.org/10.1002/we.1739">https://doi.org/10.1002/we.1612</a> and content of this study. The sentence "... are assumed based on empirical values commonly used for reference turbines by DTU." is added in line 289f.

**Referee 3:**

Comment 1: Please include a discussion of who should be interested in this research.

• A corresponding sentence was added in line 87f.

Comment 2: Figure 1 identifies, and the text briefly describes the load time series processing paths. However, none of the nomenclature (i.e. abbreviations, variables, symbols, subscripts) in Figure 1 are described. To aid the reader, please define all terms used in Figure 1 in the Section 2 text or include a nomenclature section at the beginning of the manuscript.

All nomenclature is described in the sections in chapter 2.2-2.7. Therefore a
corresponding sentence is added (line 92f). Also for clarification an
inconsistency in nomenclature was corrected changing Load Amplitude from "A"

to " $L_A$ " and mean load from "M" to " $L_M$ " in Eqs. 5a, 5b, 5c, 6, 7 and 11 and an explanation is added in line 161f.

Comment 3: The manuscript states " ... and  $p_{ws,j}$ ,  $p_{ws,j}$ ,  $p_{DLC,j}$  ... " The second instance of  $p_{ws,j}$  should be  $p_{yaw,j}$ .

• Line 204 was corrected accordingly.

Comment 4: The manuscript states "... only two cases when they are proportional: (I) When ... or (case 2.1) When ...". Please clarify if "case 2.1" refers to the processing path 2.1 or is it the second case of proportionality and therefore should be labelled as "(II)".

• Line 218 It is the second case and the mistaken label is changed to (II).

Comment 5: The test case assumes the blade material to be only uniax CFRP and GFRP. However, the IEA 22MW Reference Wind Turbine (RWT) blade contains uniax CFRP and uniax, biax, and triax GFRP, as well as medium density foam. Please clarify how the correct blade cross sectional stiffness was maintained using only uniax CFRP and GFRP. Also, while the IEA 22MW RWT does not include adhesive joints, please comment on the lack of adhesive joints in the test case and proposed methodology to define fatigue test loads.

 For the aero-elastic simulations the cross sectionals properties of the blade were derived taking all materials contained in the RWT into account. Only for the fatigue evaluation (i.e. mean load correction and accumulation) of the load time series in this study the material assumptions were simplified. The manuscript is adapted in line 266-275 to clarify this. Further Methodologies are out of the scope of this work.

Comment 7: Please identify and describe the optimization methodology executed.

• For the optimization the Nelder-Mead algorithm, as implemented in the SciPy Python package. A corresponding citation is added to the manuscript in line 340.

Comment 8: The manuscript statement "... evaluated by evaluation the test loads ..." is not grammatically correct. Please correct. Possible corrections are "... evaluated by evaluation of the test loads ...", "... evaluated by evaluating the test loads ...", or rewriting the statement to "... determined by evaluation the test loads ...".

• The manuscript is changed to "...determined by evaluating the test loads..." in line 346.

Comment 9: The future research areas should also discuss the limitations of the proposed methodologies and future research needed to address issues resulting from the combination of research assumptions (in Section 2.1), material simplification (in Section 3) and assumptions of negligibility (in Appendices A1 and A2).

 A revised sentence considering future fields of possible research was added in line 405-408.

Comment 10: The manuscript states "... zero in the in extensional ...". The second "in" is not required.

• Line 442 was corrected accordingly.

**Enhanced approach to match damage-equivalent loads in rotor blade fatigue testing**

David Melcher1, Sergei Semenov2, Peter Berring2, Kim Branner2, and Enno Petersen1

[revised manuscript text omitted]

**2 Data processing methods to derive target loads**

As every Original Equipment Manufacturer (OEM) has different procedures to derive their target loads for rotor blade fatigue tests and there is no exact procedure in the standards described, here a conventional procedure is assumed. The different processing procedures which are described in this work are visualized as a flow diagram in Fig. 1. All processing steps and the

corresponding nomenclature shown in Fig. 1 are described in Sect. 2.2 to Sect. 2.7. Processing path 0 is the minimum procedure necessary to derive damage-equivalent loads (DEL) in the main directions of the blade. However, it is not recommended as is it does not take into account any stiffness properties of the blade sections. Processing path 1, resulting in  $M_{\beta,DEL}$ , represents the assumed conventional procedure. The results from processing paths 3.1 and 3.2 are used here as a reference ease because they best represent the actual material fatigue behavior. Processing path 2.1 describes the procedure proposed by Castro et al. (2021b), and processing path 2.2 describes the enhanced approach proposed in this work.

To evaluate these target load distribution for a distributions for rotor blade fatigue test testing the procedure described in the following sections is followed.

**2.1 Underlying assumptions**

110

115

To follow the industrial standards, certain safety factors need to be considered for must be considered in the design of the fatigue tests, which are omitted in this work for simplification.

All described procedures procedures described in this work follow certain simplifying assumptions, which are listed below.

15 If any of these assumptions would be considered non-applicable, the methods described in this work here would need to be adjusted accordingly:

- 1. Validity of Timoshenko (1921) beam theory with small deformations, i.e., neglectable negligible in-plane warping of the blade sections, neglectable blade sections and negligible Brazier effect (Brazier, 1927). Otherwise, the sectional stiffness components would become dependent on these deformations -(e.g., Brazier effect reduces outer dimensions, which in turn reduces the bending stiffness).
- 2. Only longitudinal strain is considered, i.e., shear, through-thickness, and transverse strains are assumed negligible. See appendix Appendix A1 for more details.
- 3. Longitudinal strain is only affected affected only by bending moments and axial force, i.e., influence of the the influence of torque or shear loads (e.g., via Bend-twist coupling) are bend-twist coupling) is assumed negligible. See appendix A2 for more Appendix A2 for details.
- 4. Prismatic beam response is assumed, i.e., tapering or other longitudinal changes , e.g., ply drops) do not affect the strains.
- 5. Stress and strain are assumed proportional.
- 6. Material fatigue damage adheres to linear damage accumulation (Palmgren, 1924; Miner, 1945).
- 7. Material fatigue damage adheres to a linear stress-life relationship, i.e., The the Basquin curve exponent (Basquin, 1910) is independent of load levels and cycle numberslevel and cycle number.

Figure 1. Flow diagram of procedures for processing of load time series resulting in alternative DELs.

**Figure 2.** Relations between local reference coordinate system (black), elastic center EC coordinate system in principal orientation (red) and point P on the blade surface at angle  $\alpha_P$  (blue) with corresponding variables and loads.

**2.2 Load simulation**

First, the DLCs which are to be considered are chosen (e.g., from the standard IEC 61400-1:2019. And corresponding IEC 61400-1:2019. Corresponding aero-elastic turbine simulations are performed, resulting in set sets of time series f(t) for load distributions along the blade length, i.e., sectional bending moments  $M_x(t)$ ,  $M_y(t)$ , and longitudinal force  $F_z(t)$ . These loads are derived for a local reference coordinate system where in which the x-y-plane of the section is perpendicular to the blade's beam axis, which includes following the orientation of any pre-bend or sweep curvature of the blade. Also the reference line (e.g., pre-bend). The coordinate system's position and orientation in which the loads are reported in, need to must follow the blade deformation during simulation. Otherwise, the longitudinal z-axis would not be perpendicular to the cross-section plane, anymore and the subsequently used and the beam theory formulas used subsequently would not be valid. Here, it is assumed that, in the undeformed state, the projection of this local coordinate system's x-axis to onto the blade's root section is parallel to the global lead-lag direction of the blade for any section and does not follow the blade's twist angle. Following the twist angle or other orientations are also possible procedures also possible. For the same sections along the blade, for which the loads are derived, the following properties are computed (see Fig. 2):

- Coordinates of the elastic center (EC), i.e., the point where a force applied normal to the cross section will produce produces no bending curvatures:  $x_{EC}$ ,  $y_{EC}$
- Angle of principal stiffness axes orientation:  $\theta_{pa}$  (also known as structural pitch)
- Principal bending stiffnesses about the  $x^e$  and  $y^e$  axis relative to elastic center axes relative to EC:  $EI_{x^e}$ ,  $EI_{y^e}$
- Axial stiffness: EA

135

**140 2.3 Transformation of load time series**

The load time series are transformed to the elastic center EC and into the principal axes orientation of the corresponding section of the blade according to Eq. 1:

$$\begin{bmatrix} M_{x^{e}} \\ M_{y^{e}} \\ F_{z^{e}} \end{bmatrix} = \begin{bmatrix} \cos \theta_{pa} & \sin \theta_{pa} & 0 \\ -\sin \theta_{pa} & \cos \theta_{pa} & 0 \\ 0 & 0 & 1 \end{bmatrix} \cdot \begin{bmatrix} 1 & 0 & -y_{EC} \\ 0 & 1 & x_{EC} \\ 0 & 0 & 1 \end{bmatrix} \cdot \begin{bmatrix} M_{x} \\ M_{y} \\ F_{z} \end{bmatrix}$$
(1)

This load transformation is necessary as the following equations for strain are only valid for the elastic center EC in principal orientation (see Appendix A2).

From this, the longitudinal strain at any given point of interest P within the corresponding blade section (see Fig. 2) can be computed:

$$\varepsilon_z(\mathbf{P}) = \frac{y_{\mathbf{P}}^{\mathbf{e}}}{EI_{x^e}} \cdot M_{x^e} - \frac{x_{\mathbf{P}}^{\mathbf{e}}}{EI_{y^e}} \cdot M_{y^e} + \frac{F_{z^e}}{EA}$$
(2)

Assuming the longitudinal force contribution is negligible compared to the bending moment contribution, i.e.,  $\frac{F_{z^o}}{EA} \approx 0$ , and by utilizing utilizing the distance  $r_P$  from the elastic center to P EC to P and its angle  $\alpha_P$ , the strain can be written as:

$$\varepsilon_{z,M}(\mathbf{P}) = \frac{r_{\mathbf{P}} \cdot \sin \alpha_{\mathbf{P}}}{EI_{x^{\mathbf{e}}}} \cdot M_{x^{\mathbf{e}}} - \frac{r_{\mathbf{P}} \cdot \cos \alpha_{\mathbf{P}}}{EI_{y^{\mathbf{e}}}} \cdot M_{y^{\mathbf{e}}}$$
(3)

In this work, the strain time series  $\varepsilon_z(P,t)$  are used as reference  $\frac{1}{2}$  because they are assumed to be the most realistic representation of the materials fatigue behavior.

The bending moment  $M_{\beta}$  perpendicular to the direction of  $r_{\rm P}$ , which is usually assumed to contribute most to the strain  $\varepsilon_{z,M}$ , can be calculated by coordinate transformation:

$$M_{\beta} = \sin \alpha_{\mathbf{P}} \cdot M_{x^{\mathbf{e}}} - \cos \alpha_{\mathbf{P}} \cdot M_{y^{\mathbf{e}}} \tag{4}$$

In the assumed conventional procedure, only this bending moment  $M_{\beta}$  is considered, especially only particularly for the global blade main directions, i.e., flapwise and lead-lag, which, under the assumed coordinate system orientation, correspond to  $\alpha_{P,f} = -\theta_{pa}$  and  $\alpha_{P,l} = -\theta_{pa} + 90^{\circ}$ , respectively.

**160 2.4 Rainflow counting**

165

Any given load (bending moment, strain, etc.) must be further processed. In the following, L is used as a placeholder for any available load measure. To accumulate the load time series from the simulation L(t) from simulations into corresponding DELs, the time series are converted via the Rainflow counting algorithm (ASTM E1049-85) into a list of occurring load amplitudes  $A_i L_{A,i}$  with corresponding mean loads  $M_i L_{M,i}$  and cycle numbers  $n_i$ . This list can be compressed further into so-called Markov matrices by sorting the loads into discrete intervals  $\frac{1}{2}$ , i.e., binningthem (binning). Here, no binning was applied.

**2.5 Mean load correction**

170

175

180

185

Only after Rainflow counting and before the next processing step the can mean load correction (MLC) can be applied. This step is necessary to account accounts for the effect of the mean load on material fatigue. This It entails changing a load amplitude  $A_i L_{A,i}$ , which corresponds to a specific mean load  $M_i L_{M,i}$ , to a corrected load amplitude  $A_{i,MLC} L_{A,MLC,i}$ , which in turn corresponds to the mean load  $M_{i,MLC} = 0$ . This corrected amplitude is computed such that it contributes the same material fatigue damage as the original  $A_i L_{A,i} - M_i$  pair  $L_{M,i}$  pair. This correction requires the use of constant-life diagrams (CLDCLDs), which are material specific.

The simplest form of this is a liner linear symmetric CLD (also known as the Goodman or Goodman-Haigh diagram), which only requires one ultimate load U— $L_U$  and assumes symmetric behaviour behavior in tension and compression. For this, the MLC can be performed according to Eq. 5a:

$$\underline{A_{i,\text{MLC}}} \underline{L_{\text{A},\text{MLC},i}} = \underline{A_{i}} \underline{L_{\text{A},i}} \cdot \underline{U} \underbrace{L_{\text{U}}}_{U-|M_{i}|} \underline{L_{\text{U}}-|L_{\text{M},i}|}$$
(5a)

As most composite materials have different properties in tension and compression, a shifted Goodman diagram is proposed in DNV ST-0376:2024. This used uses different ultimate loads for tension  $U_{\text{t}}$  and compression  $U_{\text{c}}L_{\text{Ut}}$  and compression  $L_{\text{Uc}}$ , which results in Eq. 5b for the MLC:

$$A_{i,\text{MLC}}\underbrace{L_{\text{A,MLC},i}} = A_{i}\underbrace{L_{\text{A,}i}} \cdot \underbrace{\frac{U_{\text{avg}} - |U_{\text{mid}}|}{U_{\text{avg}} - |M_{i} - U_{\text{mid}}|}}_{\underbrace{L_{\text{U,avg}} - |L_{\text{M,}i} - L_{\text{U,mid}}|}_{\underbrace{L_{\text{U,avg}} - |L_{\text{M,}i} - L_{\text{U,mid}}|}}_{\underbrace{U_{\text{avg}} - |L_{\text{M,}i} - L_{\text{U,mid}}|}_{\underbrace{U_{\text{LU,avg}}}}} \quad \text{with} \quad \underbrace{U_{\text{avg}} L_{\text{U,avg}}}_{\underbrace{L_{\text{U,avg}}} = \underbrace{\frac{|U_{\text{t}} - U_{\text{c}}|}{2}}_{\underbrace{L_{\text{U,t}} - L_{\text{Uc}}|}_{\underbrace{L_{\text{U,t}}}}}, \quad \underbrace{U_{\text{mid}} L_{\text{U,mid}}}_{\underbrace{L_{\text{U,mid}}}} = \underbrace{\frac{U_{\text{t}} + U_{\text{c}}}{2}}_{\underbrace{L_{\text{U,t}}}}$$

$$(5b)$$

Also more More complex CLDs as proposed by Sutherland and Mandell (2005) can also be employed. In that case, the implementation of Eq. 5 in the load evaluation would need to be replaced by corresponding methods.

Since the required material properties for MLC are only available for stress or strain data, this correction is not possible for bending moments. Therefore, when employing the conventional bending moment based bending-moment-based approach, the impact of the mean load cannot be taken into account and has to must be neglected, resulting in Eq.5e:-5c:

$$A_{i,\text{MLC}}L_{A,\text{MLC},i} = A_{i}L_{A,i} \tag{5c}$$

**2.6 Linear damage accumulation**

After MLC, the corrected load amplitudes  $A_{i,\text{MLC}}$  for each simulation are accumulated into a single DEL amplitude  $A_{\text{DEL}}$  with an arbitrary cycle number  $N_{\text{DEL}}$ , using linear damage accumulation according to Palmgren (1924) and Miner (1945), assuming a linear stress-life relationship according to Basquin (1910):

$$A\underline{L}_{DEL} = \left(\frac{\sum_{i} (n_i \cdot (A_{i,MLC})^m)}{n_{DEL}} \sum_{i} (n_i \cdot (L_{A,MLC,i})^m)}{n_{DEL}}\right)^{\frac{1}{m}}$$
(6)

where m denotes the negative inverse Basquin curve exponent of the material under investigation.

There are several approaches how to define this arbitrary amount for defining this arbitrary number of cycles  $n_{\text{DEL}}$ . Some research suggested to use the dominating using the dominant frequency of the blade if it is contained in contained in the load spectrum or otherwise the zero/mean crossing frequency (Veers, 1982). Another approach was to pick up is to pick a frequency of 1 Hz, which was representative for a turbine size (Madsen et al., 1984). The latter approach was widely adopted because simulation time t in seconds is equal to the number of cycles  $n_{\text{DEL}}$ , and nowadays 1 Hz equivalent load is the commonly accepted practice, resulting in  $n_{\text{DEL}} = t$ .

After the loads for each separate simulation j are accumulated into one damage-equivalent load  $A_{\text{DEL},j}$  Local each with 200  $n_{\text{DEL},j} = 1$  according to Eq. 6, the loads from different simulations are accumulated into one total damage-equivalent load amplitude  $A_{\text{DEL},\text{total}}$  Local with a cycle number of  $N_{\text{DEL},\text{total}}$  using probabilities of occurrence as weighting factors:

$$A\widetilde{L}_{\text{DEL,total}} = \left(\frac{\sum_{j} \left(\frac{LT}{t_{j}} \cdot n_{\text{DEL},j} \cdot p_{ws,j} \cdot p_{yaw,j} \cdot p_{\text{DLC},j} \cdot (A_{\text{DEL},j})^{m}\right)}{N_{\text{DEL,total}} \cdot \sum_{j} \left(n_{ts,j} \cdot p_{ws,j} \cdot p_{yaw,j} \cdot p_{\text{DLC},j}\right)} \frac{\sum_{j} \left(\frac{LT}{t_{j}} \cdot n_{\text{DEL},j} \cdot p_{ws,j} \cdot p_{yaw,j} \cdot p_{\text{DLC},j} \cdot (L_{\text{DEL},j})^{m}\right)}{N_{\text{DEL,total}} \cdot \sum_{j} \left(n_{ts,j} \cdot p_{ws,j} \cdot p_{yaw,j} \cdot p_{\text{DLC},j}\right)} \right)$$

$$(7)$$

where LT denotes the total expected turbine design lifetime,  $t_j$  the duration of the time series,  $n_{ts,j}$  the number of turbulence seeds (i.e., the number of simulations with the same conditions), and  $p_{ws,j}$ ,  $p_{ws,j}p_{ww,j}$ ,  $p_{DLC,j}$  the probabilities of the simulations simulation's wind speed, yaw angles, and design load case (DLC), respectively. If further variables are differentiated with more simulations, the probabilities need to be adapted accordingly. As each simulation contains three blades, the loads from each blade can be considered as separate simulation runs. This effectively triples the number of turbulence seeds  $n_{ts,j}$ ; if the if loads for all three blades are evaluated and accumulated.

Note, this damage accumulation is only valid for a linear stress-life (assumption 7 in section relationship (Assumption 7 in Sect. 2.1). To consider more complex fatigue behaviour behavior (e.g., Stüssi (1955); Rosemeier and Antoniou (2022)), the damage accumulation (Eq. 6 and 7) needs to must be adjusted accordingly.

The resulting load DELs can then be used as target loads for blade fatigue testing. Depending on the scope of the fatigue test, the amount number of investigated angles  $\alpha_P$  and blade sections, has to must be chosen correspondingly. The fatigue tests then have to be designed to match or exceed these target loads.

**215 2.7 Methods for the enhanced procedure**

195

205

210

220

From EqEqs. 3 and 4, it can be seen that the strain and the swept bending moment are generally not proportional,  $\varepsilon_{z,M} \not < M_{\beta}$ . There are only two cases when they are proportional: (I) When when the two principal stiffnesses of the section are equal  $(EI_{x^e} = EI_{y^e})$ , which is usually only the case at the cylindrical root of the rotor blade, or (ease 2.1) When II) when the position of interest P is on the principal axes, i.e.,  $\alpha_P = 0^\circ; \pm 90^\circ; 180^\circ$ . As the conventional target loads are usually based on these bending moments and material fatigue damage is based on stresses or strains, this non-proportionality leads to the discrepancies between the fatigue loads in blade fatigue testing and material fatigue damage which arise arising from the design loads.

Further discrepancies can arise if the moments are converted into DELs while omitting the load transformation into the elastic center EC (see path 0 in Fig. 1). The impact of this is outside the scope of this work as it is highly dependent on the arbitrary position of the used coordinate systems.

225

To mitigate the problem of non-proportionality of bending moments and strain, Castro et al. (2021b, 2024) proposed the modified bending moment  $M'_{\beta}$  to be used as basis for the the basis for target loads instead of the regular bending moments  $M_{\beta}$  (see path 2.1 in Fig. 1). In this work,  $M'_{\beta}$  has been slightly modified compared to Castro et al. (2021b) to eloser more closely represent strain values:

230
$$M_{\beta}' = \sin \alpha_{\mathbf{P}} \cdot M_{x^{\mathbf{e}}} - \cos \alpha_{\mathbf{P}} \cdot \frac{EI_{x^{\mathbf{e}}}}{EI_{y^{\mathbf{e}}}} \cdot M_{y^{\mathbf{e}}}$$
 (8)

Translating the loads into  $M'_{\beta}$  for the test design instead of transforming the data directly into strains has the benefit that the data required for the translation does do not contain sensitive design data of the bladeblade design data, which helps with data transfer between OEM and test center, as highlighted by Castro et al. (2021b). With information on geometry and stiffness properties,  $M'_{\beta}$  can be transformed into the corresponding strain:

$$\quad \varepsilon_{z,M} = \frac{r_{\rm P}}{EI_{r^{\rm e}}} \cdot M_{\beta}' \tag{9}$$

This transformation is only valid if the assumption of the contribution of the longitudinal force to the strain being that the longitudinal force contribution to strain is negligible holds. Therefore, the impact of this assumption is investigated in section Sect. 3.1.

For the MLC of  $M'_{\beta}$ , Castro et al. (2021b) proposed an approach based on the symmetric Goodman-Haigh diagram, but without the use of material databut rather; instead, unspecified ultimate loads derived from experience of the test institution's experience were used. Also, the symmetry which only requires angles  $\alpha_P = 0^{\circ}...180^{\circ}$  does not hold anymore once the MLC is applied, and angles  $\alpha_P = -180^{\circ}...180^{\circ}$  are required. As Eq. 9 allows for simple conversion between strain  $\varepsilon_{z,M}$  and  $M'_{\beta}$ , this enables the use of MLC material-based MLC can be used in the  $M'_{\beta}$  -domain with material based data by converting the CLD-data domain by converting CLD data into the  $M'_{\beta}$  -domain (see path 2.2 in domain (see Fig. 1, path 2.2). For example, to enable Eq. 5b, the ultimate tension and compression loads need to must be evaluated:

$$M'_{\beta,\mathrm{Ut}} = \frac{EI_{x^{\mathrm{e}}}}{r_{\mathrm{P}}} \cdot \varepsilon_{z,\mathrm{Ut}}, \qquad M'_{\beta,\mathrm{Uc}} = \frac{EI_{x^{\mathrm{e}}}}{r_{\mathrm{P}}} \cdot \varepsilon_{z,\mathrm{Uc}}$$

$$\tag{10}$$

Though However, this requires the derivation of individual CLD data for every position of interest along the blade. Also,  $M'_{\beta}$  after MLC is not no longer independent of  $r_{\rm P}$  anymore and is only valid for the position for which the corresponding CLD data were are derived. If multiple positions along the  $\alpha_{\rm P}$  direction with different  $r_{\rm P}$  are of interest, multiple CLDs have to must be considered for the same  $M'_{\beta}$ . But the benefit of confidentiality still holds as The confidentiality benefit still holds because no direct material data need to be known for this disclosed for MLC.

**3 Case studies - Investigation of assumptions and methods**

255

260

265

270

275

To demonstrate the differences between the methods described above, the 138 m long reference blades of the IEA 22 MW offshore reference wind turbine (Zahle et al., 2024) are used here as a test case. Load time series are generated from aero-elastic simulations using HAWC2 (Larsen and Hansen, 2024) of for different design load cases of this turbine. These The simulations cover wind speeds from 3-25 m s-1 in 1 m s-1 steps, with yaw misalignment of 0°, 8°, and 352°, and six turbulence seeds each, while considering all three blades ( $n_{ts} = 18$ ). These simulations represent the power production design load case with the normal turbulence model (DLC 1.2) according to IEC 61400-1:2019. For the design and certification, further load cases of the turbine concerning power loss during production (DLC 2.4), start-up (DLC 3.1), shut-down (DLC 4.1), and parked (DLC 6.4) need to be considered. However, the standard does not specify individual contributions of these load cases in the to turbine lifetime and leave leaves this decision to the designer. All service and emergency load cases are design dependent, whereas DLC 1.2 mainly depends on probability of wind speed distribution the probability distribution of wind speeds between cut-in and cut-out speeds and is considered to occur approximately 95 % of the turbine lifetime (Gözcü and Verelst, 2020). Therefore, for simplification, the other DLCs are not considered in this study. This results in a total number of 1242 load time series of t = 600 s each with a resolution of 0.01 s. These load time series are evaluated computed at 49 sections along the blade span, for which the cross-sectional stiffness properties are known, derived from the structural design described by Zahle et al. (2024) and evaluated using the BECAS cross-sectional tool (Blasques and Stolpe, 2012).

These time series are evaluated as described above.

For simplification of the test case the material For the fatigue evaluation of the load time series (i.e., mean load correction and linear damage accumulation) in this case study, the material properties used are simplified. The materials of the blade is are assumed to be uniax carbon fibre reinforced uniaxial carbon fiber-reinforced polymer (CFRP) on the spar caps and uniax glass fibre reinforced uniaxial glass fiber-reinforced polymer (GFRP) everywhere else in the bladeelsewhere. The fatigue evaluation of biaxial and triaxial GFRP, as well as foam and adhesive material, are omitted in this study. However, the proposed method would allow for any number of materials to be considered, if the corresponding material properties are available. The assumed material parameters for the evaluation fatigue evaluation towards fiber-fracture are listed in Table 1. The MLC in this study is performed utilizing Eq. 5b.

Table 1. Fatigue properties of materials (IEC 61400-5:2020; Zahle et al., 2024; Camarena et al., 2022).

| Material | m  | $arepsilon_{z,	ext{Ut}}$ | $arepsilon_{z,	ext{Uc}}$ |
|----------|----|--------------------------|--------------------------|
| CFRP     | 14 | 0.0160                   | -0.0110                  |
| GFRP     | 10 | 0.0255                   | -0.0148                  |

In this example, the local reference coordinate systems of the sections have the same orientation as the global blade coordinate system, only following the blade's pre-bend and deformation. The sweep angle  $\varphi$  in the following result plots is defined as  $\varphi = \alpha_P + \theta_{pa}$  measured from the elastic center. EC.

Here, the loads were evaluated as described above for all sweep angles  $\varphi$  between -180° and 180° in 0.5° steps. For each angle, highest the largest  $r_P$ , i.e., the most outer outermost shape of the blade, was used as this has the highest strain and is assumed to be the most critical.

The loads for each separate simulation j are accumulated individually according to Eq. 6 with  $n_{\text{DEL},j} = 1$ , As only the DLC 1.2 is used in this study, Eq. 7 is adjusted here as shown in Eq. 11 to combine the different simulation results into one damage-equivalent load amplitudes amplitude:

$$A\underline{L}_{\text{DEL,total}} = \left(\frac{\sum_{j} \left(\frac{LT}{t_{j}} \cdot p_{ws,j} \cdot p_{yaw,j} \cdot (A_{\text{DEL},j})^{m}\right)}{N_{\text{DEL,total}} \cdot 0.95 \cdot \sum_{j} \left(n_{ts,j} \cdot p_{ws,j} \cdot p_{yaw,j}\right)} \frac{\sum_{j} \left(\frac{LT}{t_{j}} \cdot p_{ws,j} \cdot p_{yaw,j} \cdot (L_{\text{DEL},j})^{m}\right)}{N_{\text{DEL,total}} \cdot 0.95 \cdot \sum_{j} \left(n_{ts,j} \cdot p_{ws,j} \cdot p_{yaw,j}\right)} \right)^{\frac{1}{m}}$$

$$(11)$$

with a turbine lifetime of LT=20 years and  $N_{\rm DEL,total}=2$  million. For the wind speeds speed probabilities  $p_{ws}$ , the a Weibull distribution with a shape parameter of 2 and a scale parameter of 11.28 is used. For the yaw angle probabilities, of  $p_{yaw}(0^{\circ})=0.5$ ,  $p_{yaw}(8^{\circ})=0.25$ , and  $p_{yaw}(352^{\circ})=0.25$  are used assumed based on empirical values commonly used for reference turbines by DTU. In the following, the impact on the results of a few specific of several optional components of the evaluation procedure are on the results is investigated.

**3.1 Impact of longitudinal force contribution**

285

290

295

300

305

The first study investigates the assumption of negligibility of that the longitudinal force is negligible. Therefore, the resulting strain DELs including MLC without the longitudinal force strain  $\varepsilon_{z,M,\text{DEL,MLC}}$  contribution contribution.  $\varepsilon_{z,M,\text{DEL,MLC}}$  and with it,  $\varepsilon_{z,\text{DEL,MLC}}$ , are compared. These measures are evaluated in the exact same way as shown exactly as shown in Fig. 1 (path 3.2) with the only difference of utilizing Eq. 2-3 or Eq. 32, respectively. The relative difference differences between them are shown in Fig. 3, where they are projected on the blade geometry (left) and plotted over blade span and anglewith the . The trailing edge (TE), the leading edge (LE), and the boundaries of the spar caps on the suction side (SS-SC) and on-pressure side (PS-SC) are marked for reference. The results show that considering the influence of the longitudinal force, compared to neglecting it, raises the accumulated DELs on the suction side by max. a maximum of 1.8 % and lowers it by min. them by a minimum of -1.8 % on the pressure side, especially close to the root. This deviation is considered negligible deemed small enough to be neglected and confirms the assumption that the longitudinal force can be neglected does not need to be considered in the fatigue test target loads. For cases when the longitudinal force should be considered anyway, targets loads can be evaluated based on strains first (Fig. 1, path 3.2 using Eq. 2) and then be translated into  $M'_{3,\text{DFL,MLC}}$  using Eq. 8.

**3.2 Impact of mean load correction**

The next study investigates the impact of the MLC on the accumulated DELs. Therefore, the DELs are evaluated with and without MLC, utilizing Eq. 5e 5b and Eq. 5b (path 3.1 and 3.2 in 5c (Fig. 1 paths 2.2 and 2.1), respectively. The relative differences between the modified bending moment DELs are shown in Fig. 4. Due to their proportionality, the same

Figure 3. Distribution of relative difference between damage-equivalent accumulated longitudinal strain amplitude (incl. MLC) with  $(\varepsilon_{z,DEL,MLC}; Fig. 1, path 3.2, using Eq. 2)$  and without  $(\varepsilon_{z,M,DEL,MLC}; Fig. 1, path 3.2, using Eq. 3)$  the consideration of longitudinal force contribution.

differences are found for the strain, i.e.,  $\frac{\varepsilon_{z,M,\text{DEL,MLC}}}{\varepsilon_{z,M,\text{DEL}}} = \frac{M_{\beta,\text{DEL,MLC}}''}{M_{\beta,\text{DEL}}''} \underbrace{\frac{\varepsilon_{z,M,\text{DEL,MLC}}}{\varepsilon_{z,M,\text{DEL,MLC}}}}_{\varepsilon_{z,M,\text{DEL,MLC}}} \underbrace{\frac{M_{\beta,\text{DEL,MLC}}''}{\varphi_{z,M,\text{DEL,MLC}}}}_{\varphi_{z,M,\text{DEL,MLC}}} \underbrace{\frac{M_{\beta,\text{DEL,MLC}}''}{\varphi_{z,M,\text{DEL,MLC}}}}_{\varphi_{z,M,\text{DEL,MLC}}} \underbrace{\frac{M_{\beta,\text{DEL,MLC}}''}{\varphi_{z,M,\text{DEL,MLC}}}}_{\varphi_{z,M,\text{DEL,MLC}}} \underbrace{\frac{M_{\beta,\text{DEL,MLC}}''}{\varphi_{z,M,\text{DEL,MLC}}}}_{\varphi_{z,M,\text{DEL,MLC}}} \underbrace{\frac{M_{\beta,\text{DEL,MLC}}''}{\varphi_{z,M,\text{DEL,MLC}}}}_{\varphi_{z,M,\text{DEL,MLC}}} \underbrace{\frac{M_{\beta,\text{DEL,MLC}}''}{\varphi_{z,M,\text{DEL,MLC}}}}_{\varphi_{z,M,\text{DEL,MLC}}} \underbrace{\frac{M_{\beta,\text{DEL,MLC}}''}{\varphi_{z,M,\text{DEL,MLC}}}}_{\varphi_{z,M,\text{DEL,MLC}}} \underbrace{\frac{M_{\beta,\text{DEL,MLC}}''}{\varphi_{z,M,\text{DEL,MLC}}}}_{\varphi_{z,M,\text{DEL,MLC}}} \underbrace{\frac{M_{\beta,\text{DEL,MLC}}''}{\varphi_{z,M,\text{DEL,MLC}}}}_{\varphi_{z,M,\text{DEL,MLC}}} \underbrace{\frac{M_{\beta,\text{DEL,MLC}}''}{\varphi_{z,M,\text{DEL,MLC}}}}_{\varphi_{z,M,\text{DEL,MLC}}}$ the DELs along the LE and TE are not affected by the MLC. However, on the SS-panels, the MLC raises the DELs by up to 7.5 %, and on the whole SS-SC between 20 m and 100 m it ranges from about 8 % to up to 10.4 % around the 80 m span. On the PS, up to the 110 m span, the DELs are lowered by at least -3%, with the lowest of -6.1% on the spar cap around the 25 m span. These deviation deviations are considered significant, and especially the increased load especially confirms the necessity of MLC. Using the conventional methods without MLC to define the target loads would therefore lead to an insufficiently tested SS-SC: A a 9.4 % decrease (opposite to a 10.4 % increase) of DELs leads to a 75 % decrease of in applied fatigue damage (with m=14), which is missing compared to the predicted fatigue damage from the time series with MLC. This would lead to a fatigue test confirming only 25 % of the intended design fatigue life. The 10.4 % discrepancy shown in this study may also be exceeded when using different material properties or different CLD-formulations.

**Impact of modified bending moment $M'_{\beta}$**

310

315

320

To investigate the impact of using the modified bending moments  $M'_{\beta}$  instead of the regular bending moments  $M_{\beta}$  for defining target loads for fatigue testing, these measures cannot simply be be simply compared by values ; as because their formulations are inherently different. Hence, for comparison, multiple sets of simplified uniaxial fatigue test loads for flapwise and lead-lag are computed here. These are designed to satisfy the different target loads respectively, i.e., to match or exceed the 325 (i.e.,  $\varphi = 180^{\circ}, 0^{\circ}, 90^{\circ}, -90^{\circ}$ ) are considered as targets. In conventional fatigue test designs the blade is fixed at the root with the blade axis horizontally and with horizontal and with the suction side facing downwards. Therefore, the blade's self-weight generates tension in the pressure side and compression in the suction side, which is assumed as the mean load for the simplified

Figure 4. Distribution of relative difference between damage-equivalent accumulated modified bending moment amplitude with  $(M'_{\beta, \text{DEL}, \text{MLC}}; \text{Fig. 1, path 2.2})$  and without  $(M'_{\beta, \text{DEL}}; \text{Fig. 1, path 2.1})$  mean load correction.

test loads (neglecting any additional masses used in real fatigue tests, e.g., load frames). The cycle number for both tests is set to  $n_{\text{test}} = 2e6n_{\text{test}} = 2 \times 10^6$ . As test amplitude, the scaled load vector vectors resulting from the first two natural bending mode shapes of the blade are considered (assuming  $F_z = 0$ ), as fatigue tests are usually excited in resonance:

330

335

$$\boldsymbol{L}_{M} = \begin{bmatrix} M_{x} \\ M_{y} = 0 \\ F_{z} = 0 \end{bmatrix}_{\text{self weight}} \qquad \boldsymbol{L}_{A,f} = S_{f} \cdot \begin{bmatrix} M_{x} \\ M_{y} \\ F_{z} = 0 \end{bmatrix}_{\text{1st mode shape}} \qquad \boldsymbol{L}_{A,l} = S_{l} \cdot \begin{bmatrix} M_{x} \\ M_{y} \\ F_{z} = 0 \end{bmatrix}_{\text{2nd mode shape}} \tag{12}$$

where  $L_{\rm M}$  is the mean load vector, and  $L_{{\rm A},f}$  and  $L_{{\rm A},l}$  are the amplitude load vectors for the flapwise and lead-lag testtests, respectively, with the corresponding scaling factors  $S_f$  and  $S_l$ . The amplitude load vectors are scaled for both tests in such a matter, such that the accumulated load loads from both tests satisfy the target loads. Therefore, the test loads are evaluated in the same way as the field loads starting from Eq. 1. This is done for each section along the blade individually and independent from each other independently. To find the right scaling factors for each section, an optimization problem needs to be is solved:

$$\underset{S_f, S_l}{\text{minimize}} \quad \sum_{\varphi \in \Phi} \left( \underbrace{A \mathcal{L}_{\text{DEL}, \varphi, \text{test}}}_{\text{Lest}}(\boldsymbol{L}_{\text{M}}, \boldsymbol{L}_{\text{A}, f}, \boldsymbol{L}_{\text{A}, l}) - \underbrace{A \mathcal{L}_{\text{DEL}, \varphi, \text{field load}}}_{\text{DEL}, \varphi, \text{field load}} \right) \tag{13a}$$

subject to
$$L_{\text{DEL},\varphi,\text{test}}(\boldsymbol{L}_{\text{M}},\boldsymbol{L}_{\text{A},f},\boldsymbol{L}_{\text{A},l} \ge L_{\text{DEL},\varphi,\text{field load}}, \quad \varphi \in \Phi = \{180^{\circ},0^{\circ},90^{\circ},-90^{\circ}\}$$
 (13b)

The optimization problem is solved using the Nelder-Mead algorithm (Gao and Han, 2012). This results in load amplitude distribution distributions for each test. Note that these load distributions do not represent actual fatigue tests , which that could be performed in reality but rather the best case scenario where the target loads are matched as elose closely as possible along the whole blade span. This optimization is executed to generate test loads designed for three different cases (easecases 1, 2.1, and

Figure 5. Distribution of relative difference between damage-equivalent accumulated longitudinal strain amplitude (incl. MLC)  $\varepsilon_{z,M,\text{DEL},\text{MLC}}$  for test loads designed for case 1, case 2.1, and case 2.2.

2.2 corresponding to paths in Fig. 1) to match the corresponding field load data for  $A_{\text{DEL}} = M_{\beta, \text{DEL}} = M_{\beta, \text{DEL}}$  (case 1), 5  $M'_{\beta, \text{DEL}}$  (case 2.1), or  $M'_{\beta, \text{DEL,MLC}}$  (case 2.2).

From this, the impact of using these different approaches can be evaluated by evaluation determined by evaluating the test loads in terms of damage-equivalent strain amplitude  $\varepsilon_{z,\text{DEL},\text{MLC}}$ . Fig. 5 shows the difference of the test loads for case 2.1 and case 2.2, each relative to case 1. Designing the test loads for  $M'_{\beta,\text{DEL}}$  (case 2.1) compared to  $M_{\beta,\text{DEL}}$  (case 1) requires generally required higher loads to achieve the target. In the lead-lag direction(i.e., the load needs to be raised by up to 14 % around 25 m blade length, which corresponds to the maximum chord length. Towards Toward the tip outboard of 107 m, the load needs to be raised even more, though this area is usually not within the area of interest of for fatigue testing. Only in the area between 77 m and 96 m the load needs does the load need to be lowered. In the flapwise direction, the load only needs to be raised by up to 3 % close to the root around 15 m. Between 60 m and 90 m, the flapwise loads for case 1 and case 2.1 are almost identical. Designing the loads for  $M'_{\beta,\text{DEL},\text{MLC}}$  (case 2.2), i.e., considering MLC, requires even higher loads. In the lead-lag direction, the loads are similar to case 2.1 and need to be raise by up to 16 % around 25 m compared to case 1. In the flapwise direction, the load is raised by 5-8 % almost along the whole blade (8-115-8 m-115 m) compared to case 1.

350

355

This shows the assumed conventional method to design fatigue test loads (case 1) can lead to severe under-testing of the blade, as the more detailed methods (case 2.1 and 2.2) require up to 16 % higher loads.

To elaborate further on these differences, Fig. 6 shows the the fatigue damage from case 1 relative to the damage from case 2.1 and case 2.2 respectively. This damage ratio is derived from the load ratio and m for the corresponding material:  $\frac{D(1)}{D(2.1 \text{ or } 2.2)} = \left(\frac{\varepsilon_{z,\text{DEL,MLC}}(1)}{\varepsilon_{z,\text{DEL,MLC}}(2.1 \text{ or } 2.2)}\right)^{m}$ . This reveals, that the 16 % required load raise in the lead-lag direction corresponds to a fatigue damage deficit of just under 80 %. In the flapwise direction between 15m and 105m case 1 only deals produces only 33 %-43 % of the fatigue damage of case 2.2.

Figure 6. Distribution of fatigue damage ratio between case 1 and case 2.1 or case 2.2 respectively.

Figure 7. Distribution of relative difference between damage-equivalent accumulated longitudinal strain amplitude (incl. MLC)  $\varepsilon_{z,M,\text{DEL},\text{MLC}}$  for conventional approach test loads (case 1) and field loads.

Though, this method only considered the main flapwise and lead-lag blade directions, i.e.,  $\varphi = 180^{\circ}, 0^{\circ}, 90^{\circ}, -90^{\circ}$ . If other directions are also considered, the test loads compared to field loads for case 1 and case 2.2 are shown in Fig. 7 and Fig. 8, respectively. For case 1, the loads along the main directions are not matched as suggested above, but for case 2.2 the loads along the main directions are tested sufficiently. But both test scenarios show large areas , which that are loaded less from the test than from the field loads(hatched areas ). in Fig. 7 and 8). These areas may include features which should be tested, such as critical structural details or significant load transitions between design elements. This suggests uniaxial fatigue testing is not sufficient insufficient to test the whole blade and more sophisticated testing methods , (e.g., biaxial testing, ) will be required to test the whole blade sufficiently.

Figure 8. Distribution of relative difference between damage-equivalent accumulated longitudinal strain amplitude (incl. MLC)  $\varepsilon_{z,M,\text{DEL},\text{MLC}}$  for enhanced approach test loads (case 2.2) and field loads.

**4 Implementation of target loads in testing**

375

380

385

390

Any of the described accumulated loads (bottom row in Fig. 1) can be used as target loads for fatigue testing. Therefore, the test loads need to must be transformed and evaluated in the same manner as the chosen target loads. These evaluated test loads can then be checked if to confirm that they meet or exceed the corresponding target loads within the areas of interest along the blade. This load evaluation needs to be done must be performed during the fatigue test execution and also within in the test design to be able to compare enable comparison against the targets.

The conventional bending moment based bending-moment-based approach (Fig. 1, path 0) for uniaxial fatigue test execution does not require a lot of extensive processing as only the constant bending moment amplitude in the main directions with the corresponding cycle number needs to be evaluated. But However, as shown above, the error of this method can be magnificent significant. To adopt the proposed approach of this work (Fig. 1, path 2.2) for a test method with constant amplitude, the bending moments measured during testing need to be transformed and processed according to EqEqs. 1, 8, and 5, the same as the target loads with accumulation (Eq. 6) of the different sequential tests —(e.g., flapwise and lead-lag test). Live Rainflow counting with the corresponding accumulation of the test loads is only required required only for testing methods involving constantly changing load amplitudes, such as biaxial testing with an arbitrary frequency ratio.

Using strains  $\varepsilon_z$  including MLC (Fig. 1, path 3.2) as target loads will lead to the same results as the proposed approach, but it requires more potentially confidential data. In order for the testing facility to design and evaluate the test, detailed geometric data and strain-based Strain-based CLDs would need to be shared by the OEM. Castro et al. (2024) showed that by using the modified bending moment  $M'_{\beta}$  less confidential data is reduces the amount of confidential data required. Using the proposed modified bending moment  $M'_{\beta}$  including MLC (Fig. 1, path 2.2) requires different transformed CLDs for each target load, which can be provided solely for the expected load levels of the corresponding  $M'_{\beta}$  and are therefore more anonymised than the strain-based data.

**5 Conclusions**

395

400

405

This work demonstrated that conventional methods for separated separate flapwise and lead-lag fatigue test sequences and target load evaluation (Fig. 1, path 1) can lead to substantial under-testing across major areas of rotor blades, potentially compromising the intended design validation. It is proposed to employ the modified bending moment  $M'_{\beta}$  as described by Castro et al. (2021b), (Fig. 1, path 2.1), in combination with a suitable mean load correction (MLC), to achieve a (Fig. 1, path 2.2), to achieve realistic load representation. This enhanced approach can be utilized to define necessary test loads that result in sufficient fatigue damage throughout all of the blade. The findings of the case study suggest that applying this methodology can require increases in target loads of by up to 16 % for uniaxial testing compared to the conventional methods. This highlights the drawbacks of the conventional method and the importance of better representation of material fatigue behavior.

Future research should focus on practical implementations implementation of the proposed methodologies into in the processing of aero-elastic simulation results and in standardized testing practices and. It should also explore additional testing techniques, such as biaxial testing, to apply the required loads on around the whole blade and to further circumference. Furthermore, incorporating additional materials and more detailed CLDs may affect the case study results. Therefore, the proposed method should be applied to state-of-the-art industrial blades with higher-resolution material modeling to further examine the observed impacts. The impact of other simplifying assumptions used in this work (e.g., a linear stress-life relationship or the negligibility of bend-twist coupling) should also be investigated to further enhance the understanding of blade fatigue behavior.

**410 Appendix A: Load assumption details**

**A1 Relevant strain tensor components**

[revised manuscript text omitted]

Translating both the load vector  $\boldsymbol{L}$  and deformation vectors vector  $\boldsymbol{\xi}$  to the cross-sectional elastic center  $\boldsymbol{\xi}$  as reference point and rotate rotating them to the principal bending axis orientation eliminates further coupling terms  $K_{34}^{\rm e} = 0$ ,  $K_{35}^{\rm e} = 0$ :

$$\boldsymbol{L}^{e} = \boldsymbol{K}^{e} \cdot \boldsymbol{\xi}^{e} = \begin{bmatrix} F_{x^{e}} \\ F_{y^{e}} \\ F_{z} \\ M_{x^{e}} \\ M_{y^{e}} \\ M_{z^{e}} \end{bmatrix} = \begin{bmatrix} K_{11}^{e} & K_{12}^{e} & 0 & 0 & 0 & K_{16}^{e} \\ K_{12}^{e} & K_{22}^{e} & 0 & 0 & 0 & K_{26}^{e} \\ 0 & 0 & K_{33} & 0 & 0 & 0 \\ 0 & 0 & 0 & K_{44}^{e} & 0 & 0 \\ 0 & 0 & 0 & K_{55}^{e} & 0 \\ K_{16}^{e} & K_{26}^{e} & 0 & 0 & 0 & K_{66}^{e} \end{bmatrix} \cdot \begin{bmatrix} \gamma_{x^{e}} \\ \gamma_{y^{e}} \\ \varepsilon_{z^{e}} \\ \kappa_{x^{e}} \\ \kappa_{y^{e}} \\ \kappa_{z} \end{bmatrix}$$

$$(A2)$$

Inverting this reduced stiffness matrix  $K^e$  leads to a reduced compliance matrix  $C^e$ :

$$\mathbf{465} \quad \boldsymbol{\xi} = (\boldsymbol{K}^{e})^{-1} \cdot \boldsymbol{L}^{e} = \boldsymbol{C}^{e} \cdot \boldsymbol{L}^{e} = \begin{bmatrix} \gamma_{x^{e}} \\ \gamma_{y^{e}} \\ \varepsilon_{z^{e}} \\ \kappa_{x^{e}} \\ \kappa_{y^{e}} \\ \kappa_{z} \end{bmatrix} = \begin{bmatrix} C_{11}^{e} & C_{12}^{e} & 0 & 0 & 0 & C_{16}^{e} \\ C_{12}^{e} & C_{22}^{e} & 0 & 0 & 0 & C_{26}^{e} \\ 0 & 0 & \frac{1}{K_{33}} & 0 & 0 & 0 \\ 0 & 0 & 0 & \frac{1}{K_{44}^{e}} & 0 & 0 \\ 0 & 0 & 0 & \frac{1}{K_{55}^{e}} & 0 \\ C_{16}^{e} & C_{26}^{e} & 0 & 0 & 0 & C_{66}^{e} \end{bmatrix} \cdot \begin{bmatrix} F_{x^{e}} \\ F_{y^{e}} \\ F_{z} \\ M_{x^{e}} \\ M_{y^{e}} \\ M_{z^{e}} \end{bmatrix}$$

$$(A3)$$

From this, the longitudinal strain equation for all surface points P  $(x_P^e, y_P^e)$  can be derived as

$$\varepsilon_{z,P}(x_{P}^{e}, y_{P}^{e}) = \kappa_{x^{e}} \cdot y_{P}^{e} - \kappa_{y^{e}} \cdot x_{P}^{e} + \varepsilon_{z^{e}} = \frac{M_{x^{e}} \cdot y_{P}^{e}}{K_{44}^{e}} - \frac{M_{y^{e}} \cdot x_{P}^{e}}{K_{55}^{e}} + \frac{F_{z}}{K_{33}}$$
(A4)

where  $K_{33} = EA$ ,  $K_{44}^{e} = EI_{x^{e}}$ ,  $K_{55}^{e} = EI_{y^{e}}$ .

- . DM prepared the code and conducted the analysis. SS and PB supported development of the methods and preparation of the case study.

  470 KB supervised and guided the conception of the methods. EP supervised DM and acquired funding. DM prepared the manuscript with contributions from all co-authors.
  - . The authors declare that they have no conflict of interest.
- . We acknowledge the support by Frederick Zahle and Athanasios Barlas from DTU with the load simulation and generation of load time series for the reference turbine test case. We also acknowledge the support provided by from the German Federal Ministry for Economic Affairs and Energy (BMWE) within the SmarTestBlade project (03EE2037).

**References**

[revised manuscript text omitted]